# TOMCAT 🐱: Test-time Comprehensive Knowledge Accumulation for Compositional Zero-Shot Learning

**Xudong Yan** [1,2]   **Songhe Feng** [1,2] *

[1] School of Computer Science and Technology, Beijing Jiaotong University
[2] Key Laboratory of Big Data and Artificial Intelligence in
Transportation (Beijing Jiaotong University), Ministry of Education
{xud_yan, shfeng}@bjtu.edu.cn

## Abstract

Compositional Zero-Shot Learning (CZSL) aims to recognize novel attribute-object compositions based on the knowledge learned from seen ones. Existing methods suffer from performance degradation caused by the distribution shift of label space at test time, which stems from the inclusion of unseen compositions recombined from attributes and objects. To overcome the challenge, we propose a novel approach that accumulates comprehensive knowledge in both textual and visual modalities from unsupervised data to update multimodal prototypes at test time. Building on this, we further design an adaptive update weight to control the degree of prototype adjustment, enabling the model to flexibly adapt to distribution shift during testing. Moreover, a dynamic priority queue is introduced that stores high-confidence images to acquire visual knowledge from historical images for inference. Considering the semantic consistency of multimodal knowledge, we align textual and visual prototypes by multimodal collaborative representation learning. Extensive experiments indicate that our approach achieves state-of-the-art performance on four benchmark datasets under both closed-world and open-world settings. Code will be available at https://github.com/xud-yan/TOMCAT.

## 1 Introduction

Integrating familiar concepts and expanding to novel compositions stand as one of the striking hallmarks of human intelligence [9, 10, 24]. For example, even when confronted with a completely new composition `browned cheese`, humans can swiftly apprehend and recognize it by combining known atomic concepts `browned` and `cheese`. These capacities for compositionality and generalization enable us to infer a wide range of combinations from limited atomic concepts [1], and have spurred the emergence of Compositional Zero-Shot Learning (CZSL) [39, 46, 2, 26, 35]. Specifically, CZSL aims to recognize unseen attribute-object compositions based on the primitive (*i.e.*, attributes and objects) knowledge learned from seen ones [60].

Traditional CZSL approaches either project both images and attribute-object labels into a joint space to learn compositional representations [40, 36, 22], or use two classifiers to predict attributes and objects separately [50, 25, 13]. Recently, benefiting from the impressive multimodal representational abilities of large pre-trained vision-language models (VLMs) like CLIP [49], there have been further significant advancements in CZSL [42, 34, 14, 28]. These methods employ the prompt tuning technique [73], *i.e.*, replacing the hard prompt like "*a photo of* [attribute] [object]" with learnable soft prompt tokens to align images and textual composition labels, thereby fine-tuning VLMs for adaptation to the CZSL task.

---

*Corresponding author

39th Conference on Neural Information Processing Systems (NeurIPS 2025).

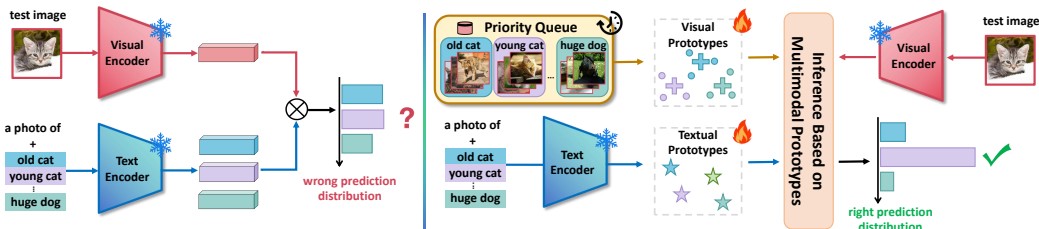

Figure 1: **At test time**, existing methods (*left*) fail to leverage test images for model updates, obtaining wrong prediction distribution caused by label space shift, whereas our TOMCAT (*right*) accumulates multimodal knowledge from unsupervised data at test time to overcome the challenge.

While these pioneering methods have achieved significant progress, they fail to address the performance degradation due to the distribution shift of label space at test time. Specifically, models are assigned to identify unseen attribute-object compositions that are absent during training. This pattern leads to a mismatch between the learned and actual test-time label distribution, bringing about inaccurate predictions and poor generalization. The key reason why the above issue has yet to be effectively addressed lies in the fact that the model parameters and class prototypes are frozen after training, preventing the model from leveraging the test data to adapt towards the new distribution.

However, in more real-world scenarios, an outstanding intelligent system should continuously accumulate knowledge and thus overcome distribution changes after deployment by utilizing the unlabeled data provided through user interactions during the testing phase. In CZSL, the following three aspects should be taken into consideration when using test-time unsupervised data to adjust the model: (1) **Accumulation**. The process of observing test-time images can be regarded as a form of knowledge accumulation, rather than a complete adaptation to unseen compositions at the cost of forgetting knowledge of seen compositions learned during training [23, 43, 58]. (2) **Knowledge comprehensiveness**. Existing methods treat compositions encoded by the text encoder of VLMs as textual-modal prototypes, and predict compositions by computing similarity between the visual features of images and prototypes. However, they overlook the potential of utilizing visual information in the historical images encountered at test time. (3) **Efficiency**. Considering the practical scenario of interaction with users, the method should be time-efficient with low latency [43, 29, 7].

To this end, we propose a **T**est-time c**OM**prehensive knowledge a**C**cumul**AT**ion (**TOMCAT**) approach for CZSL, a novel framework that leverages multimodal knowledge of unlabeled data at test time to overcome label distribution shift, as illustrated in Fig. 1. During the test phase, we keep the textual-modal prototypes frozen and progressively learn a Knowledge Accumulation Module (KAM) for prototype adjustment to bridge the label distribution gap with the continual influx of test samples. Subsequently, we determine the extent to which KAM updates the prototypes by employing a designed adaptive update weight strategy, based on similarities between the image and the prototypes. To take full advantage of the visual knowledge from previously seen images, TOMCAT maintains a dynamic priority queue to store high-confidence images for each class. Building on this, visual-modal prototypes are constructed from the images stored in the queue and are dynamically updated–similarly to the textual counterpart–by visual KAM with higher-confidence images enqueued. As testing progresses, the textual and visual prototypes are updated by the entropy minimization objective [54, 52], and are jointly used to facilitate composition recognition under the new label distribution. In addition, given the inherent semantic interdependence between multimodal knowledge, we align textual and visual prototypes by multimodal collaborative representation learning.

Notably, in practical applications where user interaction imposes high requirements on time efficiency, we keep the original model frozen during inference and optimize only the parameters of KAMs via gradient backpropagation once the prediction for a test sample is completed. Meanwhile, extensive experiments on four benchmark datasets (*i.e.*, UT-Zappos [61], MIT-States [15], C-GQA [40], and Clothing16K [68]) demonstrate that our TOMCAT outperforms the state-of-the-art by significant margins in both closed-world and open-world settings (Sec. 3.1).

In summary, the main contributions of our work are three-fold:

- We propose TOMCAT, a novel framework that accumulates multimodal knowledge from unlabeled data and updates prototypes at test time to bridge the label distribution shift. To the best of our knowledge, we are the first to leverage unsupervised data at test time to improve models in CZSL.

- TOMCAT adopts a priority queue that stores historical high-confidence images to calculate visual prototypes and adaptively updates multimodal prototypes by knowledge accumulation modules.
- Extensive experiments conducted on four benchmark datasets demonstrate that our TOMCAT achieves state-of-the-art performance in both closed-world and open-world settings.

## 2 Related Work

**Compositional Zero-Shot Learning (CZSL).** CZSL requires the model to recognize unseen compositions with the attribute and object knowledge learned from seen compositions. Previous works in CZSL can be broadly divided into two main streams. One main stream aims at learning representations of compositions by aligning images and textual labels in a shared space, and predicting compositions with the lowest distance [39, 41, 46, 40, 36, 22]. The other stream concentrates on disentangling visual representations of compositions and predicting individual primitives separately to reduce composition learning into primitive learning since both seen and unseen compositions inherently share the same attributes and objects [50, 27, 51, 56, 33, 71]. Rather than learning image-composition association from scratch, recent approaches have increasingly shifted focus to exploiting the multimodal representational capacity of VLMs (*e.g.*, CLIP [49]) for CZSL [42, 34, 14, 28, 3, 59]. For example, Troika [14] exploits multi-path paradigm and cross-modal traction modules to jointly model attributes, objects, and compositions. CDS-CZSL [28] leverages context-based diversity-driven specificity learning to prioritize specific attributes with richer information. ClusPro [47] learns multi-prototypes by within-primitive clustering and dynamically updates them. Although notable progress has been made, these methods freeze model parameters and prototypes after training, which hinders them from using test-time data for further improvement.

**Vision-Language Model (VLM).** VLMs (*e.g.*, CLIP [49] and ALIGN [16]) pre-trained on web-scale datasets have recently attracted considerable attention due to their impressive capability in aligning visual and textual modalities. Current approaches typically seek to repurpose the multimodal ability of VLMs for various downstream tasks through prompt tuning or adapter tuning. On the text side, prompt tuning is a parameter-efficient technique that replaces manually crafted prompts with a sequence of learnable soft tokens while keeping the text encoder frozen [73, 72, 74]. For example, CoOp [73] eliminates the requirement of manually crafting prompts by prompt tuning under the few-shot setting. CoCoOp [72] extends CoOp by introducing instance-wise conditional prompt learning. On the visual side, adapter tuning refers to injecting lightweight trainable modules (*e.g.*, residual adapters and attention-based adapters) into the visual encoder without fine-tuning the entire backbone [62, 8, 11, 48]. For instance, AdapterFormer [5] augments each Transformer block of the visual backbone with a parallel adapter module and fuses their outputs via element-wise addition. In this work, following CoOp [73] and AdapterFormer [5], we fine-tune CLIP using training data to obtain a simple base model for subsequent testing.

**Online Test-time Adaptation (OTTA).** OTTA refers to a practical technique where a trained model continually adapts itself by exploiting a stream of unsupervised test data–each test sample is used exactly once, and the model is required to retain and leverage the knowledge gained from earlier test images to improve its performance on later ones in an online manner [54, 55, 44, 32, 4, 6, 30]. For example, Tent [54] proposes to optimize batch normalization layer by minimizing prediction entropy on test data. SAR [44] enhances generalization ability in OTTA by eliminating partially noisy samples with high gradients and promoting the convergence of model weights to a flat minimum. CoTTA [55] is the first work to decrease the accumulation error and avoid catastrophic forgetting through using averaged pseudo-labels and retaining the knowledge of the original model to enhance long-term adaptation. Recently, activating the zero-shot capability of VLMs at test time to mitigate domain shift in downstream tasks has increasingly attracted significant research attention [52, 19, 63, 66, 64, 53]. For instance, TPT [52] develops a prompt tuning approach that learns adaptive prompts on the fly using each test sample with its augmented views. TDA [19] proposes a training-free adapter that uses a lightweight key-value cache and progressive pseudo-label refinement without backpropagation. DynaPrompt [57] adaptively selects and optimizes the relevant prompts for each test built on an online prompt buffer. TPS [53] proposes a straightforward and efficient prototype shifting approach that adjusts per-class prototypes within the embedding space. Notably, OTTA mainly focuses on addressing distribution shifts in the feature domain while we aim to resolve the challenge of label distribution shift caused by unseen compositions recombined from attributes and objects in CZSL. The CZSL task does not have access to any data of unseen compositions during training.

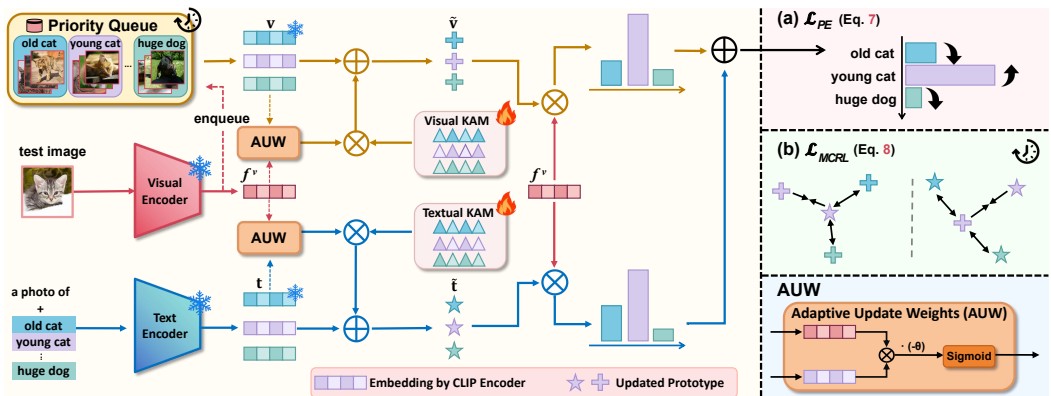

Figure 2: The overall architecture of our proposed TOMCAT at test time. The model accumulates multimodal knowledge to update prototypes to overcome the label distribution shift.

## 3 Method

### 3.1 Task Formulation

In this section, we provide a formal description of the CZSL task. Given an attribute set $A$ and an object set $O$, the composition set can be defined by the Cartesian product of $A$ and $O$, *i.e.*, $C = A \times O$. And the seen and unseen composition set $C^s$ and $C^u$ are two disjoint subsets of $C$, *i.e.*, $C^s \cap C^u = \varnothing$. During the training phase, the model learns from a seen training set $D^{tr} = \{(x, c) | x \in \mathcal{X}, c \in C^s\}$, where $\mathcal{X}$ is the image space, and $c$ is the composition label of image $x$. In the closed-world setting [46], the test composition set $C^{te}$ is defined as the union of $C^s$ and $C^u$, *i.e.*, $C^{te} = C^s \cup C^u$, where only the presupposed known composition space is considered. For the open-world setting [35], the test composition set expands to all possible attribute-object pairs, *i.e.*, $C^{te} = C$.

### 3.2 Training Phase

As a preparatory step, we train a simple CLIP-based model by adjusting only its visual and text encoders on the training set, which serves as the foundation for subsequent testing.

**Textual Representation Extraction.** CLIP treats each attribute-object composition label as a prompt-based textual input, *i.e.*, "*a photo of* [attribute] [object]", which is subsequently encoded via the text encoder $\psi$. Given a composition $c = (a, o)$, we create a learnable prompt $\mathbf{P}^c = [\mathbf{p}_1, \ldots, \mathbf{p}_l, \mathbf{w}^a, \mathbf{w}^o]$, where $\mathbf{p}_{1:l}$ represents the learnable prompt tokens. $\mathbf{w}^a$ and $\mathbf{w}^o$ are the learnable word tokens of $a$ and $o$ within the attribute and object vocabularies, respectively. Therefore, the textual representation of composition $c$ is obtained, *i.e.*, $f_c^t = \psi(\mathbf{P}^c)$.

**Visual Representation Extraction.** Given an input image $x \in \mathbb{R}^{H \times W \times 3}$, we feed it into the CLIP visual encoder $\phi$ and employ the output [CLS] token as its visual representation, *i.e.*, $f^v = \phi(x)$. Based on AdapterFormer [5], a set of learnable adapters is injected into the visual encoder of CLIP, while keeping the initial parameters frozen. Refer to Appendix A for more details.

**Training Objective.** Given the textual and visual representations, we compute the probability $p(c|x)$ and use the cross-entropy loss to align them:

$$\mathcal{L}_{tr} = -\log\, p(c|x)\,, \quad p(c|x) = \frac{\exp(\cos(f^v, f_c^t)/\tau)}{\sum_{c' \in C^s} \exp(\cos(f^v, f_{c'}^t)/\tau)}\,, \tag{1}$$

where $\tau$ denotes the temperature parameter of pre-trained CLIP and $\cos(\cdot, \cdot)$ is used to compute cosine similarity. After the training phase, a simple base model is obtained by only tuning adapters and prompt within CLIP, without introducing any additional complex modules to accelerate inference.

### 3.3 TOMCAT at Test Time

As the major novelty, we introduce TOMCAT to overcome the challenge of label distribution shift by employing unsupervised data at test time, as shown in Fig. 2. Specifically, textual and visual

composition prototypes are obtained by CLIP-encoded textual labels and a dynamic priority queue of historical images, respectively. These multimodal prototypes are then updated by Knowledge Accumulation Modules (KAM) and adaptive update weights, with the objective of minimizing prediction entropy at test time.

**Textual Prototype Construction.** In line with CLIP's principles, we treat the embeddings of both seen and unseen composition labels–encoded by the text encoder of the base model after training–as the textual-modal prototypes, *i.e.*, $\mathbf{t} = [\mathbf{t}_{c_1}, \mathbf{t}_{c_2}, \dots, \mathbf{t}_{c_{|C^{te}|}}]^\top \in \mathbb{R}^{|C^{te}| \times d}$.

**Visual Prototype Construction.** Inspired by TDA [19] and DPE [63], complementary to the text modality, we recognize that prior visual knowledge–captured from historical test images–can be leveraged to further enhance the discriminative capability of CLIP. Therefore, a dynamic priority queue is designed to selectively preserve $K$ ($K = 3$) high-confidence images, which enables TOMCAT to retain representative and reliable exemplars throughout testing. Specifically, for each seen and unseen composition $c$, we maintain a "*confidence-feature*" queue, *i.e.*, $q^c = [(h_i, f_i^v)]_{i=1}^K$, where $f_i^v \in \mathbb{R}^d$ is the visual feature of the image $x_i$ in the queue and $h_i \in \mathbb{R}$ is its prediction entropy as confidence:

$$\mathcal{H}(x_i) = - \sum_{c \in C^{te}} p(c|x_i, \tilde{\mathbf{t}}) \log p(c|x_i, \tilde{\mathbf{t}}) , \quad p(c|x_i, \tilde{\mathbf{t}}) = \frac{\exp(\cos(f_i^v, \tilde{\mathbf{t}}_c)/\tau)}{\sum_{c' \in C^{te}} \exp(\cos(f_i^v, \tilde{\mathbf{t}}_{c'})/\tau)} , \quad (2)$$

where $\tilde{\mathbf{t}}$ is the updated textual prototypes (defined below in Eq. 5). The lower prediction entropy means higher confidence, and the samples in the queue are sorted by their confidence, *i.e.*, $h_i^c \leq h_{(>i)}^c$.

The priority queue corresponding to each composition is initialized as an empty set $\varnothing$. For an incoming image $x$, we first calculate its visual feature $f^v$ and the prediction entropy $h$, and obtain its pseudo-label $c_p$ by assigning the composition with the highest predicted probability, *i.e.*, $c_p = \arg\max_{c \in C^{te}} p(c|x, \tilde{\mathbf{t}})$. If the queue for $c_p$ is not full, the pair $(h, f^v)$ is directly inserted into the queue. If the queue is full, only if the new image has a prediction entropy lower than the highest element in the queue, the highest one is replaced with $(h, f^v)$; otherwise, the queue remains unchanged. Based on the priority queue, the visual-modal prototype for each composition is computed by averaging the visual features, *i.e.*, $\mathbf{v}_c = \frac{1}{K} \sum_{i=1}^K f_i^v$. Subsequently, the visual prototypes are denoted as $\mathbf{v} = [\mathbf{v}_{c_1}, \mathbf{v}_{c_2}, \dots, \mathbf{v}_{c_{|C^{te}|}}]^\top \in \mathbb{R}^{|C^{te}| \times d}$.

**Knowledge Accumulation Module (KAM).** We aim to continuously acquire information about the new distribution from test samples, while avoiding both catastrophic forgetting of existing knowledge and excessive increases in inference latency. To this end, we introduce learnable KAMs instead of directly modifying the parameters of the base model, and employ adaptive update weights to control the extent to which the KAMs adjust the prototypes. Specifically, for multimodal prototypes $\mathbf{t}$ and $\mathbf{v}$, multimodal KAMs consist of two sets of learnable parameters, which are initialized to zero:

$$\Delta\mathbf{t} = [\Delta\mathbf{t}_{c_1}, \Delta\mathbf{t}_{c_2}, \dots, \Delta\mathbf{t}_{c_{|C^{te}|}}]^\top \in \mathbb{R}^{|C^{te}| \times d} , \quad \Delta\mathbf{v} = [\Delta\mathbf{v}_{c_1}, \Delta\mathbf{v}_{c_2}, \dots, \Delta\mathbf{v}_{c_{|C^{te}|}}]^\top \in \mathbb{R}^{|C^{te}| \times d} . \quad (3)$$

**Adaptive update weight.** Taking textual prototypes as an example, we demonstrate the process by which the prototypes are updated with a newly arrived image $x$. Specifically, the cosine similarity is computed between the visual feature $f^v$ of the image and each original prototype $\mathbf{t}_c$, based on which the adaptive update weight is calculated as follows:

$$w_c = \sigma(-\theta \cdot s_c) , \quad s_c = \cos(f^v, \mathbf{t}_c) , \quad (4)$$

where $\sigma$ denotes the Sigmoid activation function and $\theta$ is the hyperparameter that controls the degree of update. Therefore, the updated textual prototypes can be denoted as follows:

$$\tilde{\mathbf{t}} = [\tilde{\mathbf{t}}_{c_1}, \tilde{\mathbf{t}}_{c_2}, \dots, \tilde{\mathbf{t}}_{c_{|C^{te}|}}]^\top , \quad \tilde{\mathbf{t}}_c = \frac{\mathbf{t}_c + w_c \Delta\mathbf{t}_c}{||\mathbf{t}_c + w_c \Delta\mathbf{t}_c||} . \quad (5)$$

Accordingly, we can obtain the updated visual prototypes $\tilde{\mathbf{v}}$. This adaptive weighting mechanism enables more controlled updates of the prototypes, thereby avoiding treating all compositions equally regardless of familiarity. Intuitively, when the test image closely matches the original prototype, it is likely associated with a seen composition, and thus excessive adjustments should be avoided. Conversely, a large discrepancy between the test image and the prototype suggests a potentially unseen composition, permitting stronger updates to improve adaptability.

**Prediction Entropy Minimization.** Following Tip-Adapter [65], the final prediction for the input image is determined as follows:

$$p(c|x, \tilde{\mathbf{t}}, \tilde{\mathbf{v}}) = \frac{\exp\left(f^v \cdot \tilde{\mathbf{t}}_c + \alpha \mathcal{A}(f^v, \tilde{\mathbf{v}}_c)\right)}{\sum_{c' \in C^{te}} \exp\left(f^v \cdot \tilde{\mathbf{t}}_{c'} + \alpha \mathcal{A}(f^v, \tilde{\mathbf{v}}_{c'})\right)} , \quad \mathcal{A}(f^v, \tilde{\mathbf{v}}_c) = \exp\left(-\beta(1 - f^v \cdot \tilde{\mathbf{v}}_c)\right), \quad (6)$$

where $\alpha$ and $\beta$ are hyperparameters controlling multimodal balance and modulating visual-modal sharpness, respectively. At test time, minimizing prediction entropy serves as an unsupervised learning signal that encourages the model to produce more confident predictions in the target label space, thereby enhancing generalization. By reducing uncertainty, the model progressively adjusts to better aligning with the test distribution, which includes unseen compositions. Further explanations are provided in Appendix B. The loss for multimodal prediction entropy is formulated as follows:

$$\mathcal{L}_{PE} = -\sum_{c \in C^{te}} p(c|x, \tilde{\mathbf{t}}_c, \tilde{\mathbf{v}}_c) \log p(c|x, \tilde{\mathbf{t}}_c, \tilde{\mathbf{v}}_c). \quad (7)$$

**Multimodal Collaborative Representation Learning.** Considering the intrinsic semantic interdependence between multimodal knowledge, we align the textual and visual prototypes by employing multimodal collaborative representation learning. This strategy effectively facilitates the integration of both modalities, thereby enhancing the representation of both textual and visual information in a unified framework. Specifically, contrastive learning is exploited to bring the visual and textual prototypes corresponding to the same composition closer while pushing apart non-corresponding ones, as formulated below:

$$\mathcal{L}_{MCRL} = -\frac{1}{2|C^{te}|} \sum_{c \in C^{te}} \left( \log \frac{\exp\left(\cos(\tilde{\mathbf{t}}_c, \tilde{\mathbf{v}}_c)/\tau\right)}{\sum_{c' \in C^{te}} \exp\left(\cos(\tilde{\mathbf{t}}_c, \tilde{\mathbf{v}}_{c'})/\tau\right)} + \log \frac{\exp\left(\cos(\tilde{\mathbf{t}}_c, \tilde{\mathbf{v}}_c)/\tau\right)}{\sum_{c' \in C^{te}} \exp\left(\cos(\tilde{\mathbf{t}}_{c'}, \tilde{\mathbf{v}}_c)/\tau\right)} \right) .$$
$$(8)$$

**Testing Pipeline Overview.** Upon receiving a test image, we first extract its visual feature and compute the prediction entropy with the original textual prototypes. We then determine whether to update the priority queue of its pseudo-composition label based on the entropy. After performing multimodal prototype refinement by KAMs and adaptive update weights, the final inference prediction for this image is obtained. To reduce latency, the backpropagation update of KAMs is deferred until after the inference step by minimizing the total loss of multimodal prediction entropy and multimodal collaborative representation learning as follows:

$$\mathcal{L}_{TOMCAT} = \mathcal{L}_{PE} + \lambda \mathcal{L}_{MCRL} , \quad (9)$$

where $\lambda$ is the weighting coefficient of $\mathcal{L}_{MCRL}$.

# 4 Experiment

## 4.1 Experiment Setup

**Datasets.** Our proposed TOMCAT is evaluated on four commonly used datasets: UT-Zappos [61], MIT-States [15], C-GQA [40], and Clothing16K [68]. UT-Zappos and Clothing16K are two fine-grained fashion datasets, whereas MIT-States and C-GQA consist of images depicting real-world objects. In addition, prior studies [2, 68, 60] have evidenced that MIT-States [15] suffers from considerable noise, with approximately 70% of the labels being incorrect. The detailed introduction and common data splits of the four datasets are presented in Appendix C.

**Metrics.** Following the evaluation protocol of previous works [46, 35, 42], a bias term from $-\infty$ to $+\infty$ is introduced to trade off the prediction logits between seen and unseen compositions. By varying the bias term, we calculate the best **Seen** accuracy, best **Unseen** accuracy, best Harmonic Mean (**HM**) of seen and unseen accuracies [70], and the Area Under the Curve (**AUC**) drawn with seen and unseen accuracies. In the open-world setting, a post-training feasibility calibration is applied to filter out infeasible compositions within a vast search space [35].

**Implementation Details.** We implement the base model with CLIP ViT-L/14 architecture in the training phase and TOMCAT at test time in PyTorch [45] framework on a single NVIDIA RTX 3090 GPU. Refer to Appendix D for more implementation details. The source code will also be released at this website to provide all implementation details and thus facilitate reproducibility.

**Baselines.** We compare TOMCAT with recent and prominent approaches on UT-Zappos [61], MIT-States [15], and C-GQA [40], including CLIP [49], CoOp [73], CLIP-based Co-CGE [36], DFSP [34],

Table 1: Closed-world and open-world results on UT-Zappos, MIT-States, and C-GQA. The best results are displayed in **boldface**, and the second-best results are underlined. The four indicators are explained in Metrics (Sec. 4.1). In the open-world setting, we report the results of CDS-CZSL* [28] using the same post-training feasibility calibration [35] as our TOMCAT and other baselines use.

| Methods | UT-Zappos | | | | MIT-States | | | | C-GQA | | | |
|---|---|---|---|---|---|---|---|---|---|---|---|---|
| | AUC | HM | Seen | Unseen | AUC | HM | Seen | Unseen | AUC | HM | Seen | Unseen |
| **Closed-world Results** | | | | | | | | | | | | |
| CLIP [49] (*ICML'21*) | 5.0 | 15.6 | 15.8 | 49.1 | 11.0 | 26.1 | 30.2 | 46.0 | 1.4 | 8.6 | 7.5 | 25.0 |
| CoOp [73] (*IJCV'22*) | 18.8 | 34.6 | 52.1 | 49.3 | 13.5 | 29.8 | 34.4 | 47.6 | 4.4 | 17.1 | 20.5 | 26.8 |
| Co-CGE [36] (*TPAMI'22*) | 36.3 | 49.7 | 63.4 | 71.3 | 17.0 | 33.1 | 46.7 | 45.9 | 5.7 | 18.9 | 34.1 | 21.2 |
| CSP [42] (*ICLR'23*) | 33.0 | 46.6 | 64.2 | 66.2 | 19.4 | 36.3 | 46.6 | 49.9 | 6.2 | 20.5 | 28.8 | 26.8 |
| DFSP [34] (*CVPR'23*) | 36.0 | 47.2 | 66.7 | 71.7 | 20.6 | 37.3 | 46.9 | 52.0 | 10.5 | 27.1 | 38.2 | 32.0 |
| GIPCOL [59] (*WACV'24*) | 36.2 | 48.8 | 65.0 | 68.5 | 19.9 | 36.6 | 48.5 | 49.6 | 7.1 | 22.5 | 31.9 | 28.4 |
| Troika [14] (*CVPR'24*) | 41.7 | 54.6 | 66.8 | 73.8 | 22.1 | 39.3 | 49.0 | 53.0 | 12.4 | 29.4 | 41.0 | 35.7 |
| CDS-CZSL [28] (*CVPR'24*) | 39.5 | 52.7 | 63.9 | 74.8 | 22.4 | 39.2 | 50.3 | 52.9 | 11.1 | 28.1 | 38.3 | 34.2 |
| PLID [3] (*ECCV'24*) | 38.7 | 52.4 | 67.3 | 68.8 | 22.1 | 39.0 | 49.7 | 52.4 | 11.0 | 27.9 | 38.8 | 33.0 |
| IMAX [17] (*TPAMI'25*) | 40.6 | 54.2 | 69.3 | 70.7 | 21.9 | 39.1 | 48.7 | 53.8 | 12.8 | 29.8 | 39.7 | 35.8 |
| ClusPro [47] (*ICLR'25*) | 46.6 | 58.5 | 70.7 | **76.0** | **23.8** | **40.7** | 52.1 | **54.0** | 14.9 | 32.8 | 44.3 | 37.8 |
| **TOMCAT (Ours)** | **48.3** | **60.2** | **74.5** | 72.8 | 22.6 | 39.5 | 50.3 | 53.0 | **16.0** | **34.0** | **45.3** | **40.1** |
| **Open-world Results** | | | | | | | | | | | | |
| CLIP [49] (*ICML'21*) | 2.2 | 11.2 | 15.7 | 20.6 | 3.0 | 12.8 | 30.1 | 14.3 | 0.3 | 4.0 | 7.5 | 4.6 |
| CoOp [73] (*IJCV'22*) | 13.2 | 28.9 | 52.1 | 31.5 | 2.8 | 12.3 | 34.6 | 9.3 | 0.7 | 5.5 | 21.0 | 4.6 |
| Co-CGE [36] (*TPAMI'22*) | 28.4 | 45.3 | 59.9 | 56.2 | 5.6 | 17.7 | 38.1 | 20.0 | 0.9 | 5.3 | 33.2 | 3.9 |
| CSP [42] (*ICLR'23*) | 22.7 | 38.9 | 64.1 | 44.1 | 5.7 | 17.4 | 46.3 | 15.7 | 1.2 | 6.9 | 28.7 | 5.2 |
| DFSP [34] (*CVPR'23*) | 30.3 | 44.0 | 66.8 | 60.0 | 6.8 | 19.3 | 47.5 | 18.5 | 2.4 | 10.4 | 38.3 | 7.2 |
| GIPCOL [59] (*WACV'24*) | 23.5 | 40.1 | 65.0 | 45.0 | 6.3 | 17.9 | 48.5 | 16.0 | 1.3 | 7.3 | 31.6 | 5.5 |
| Troika [14] (*CVPR'24*) | 33.0 | 47.8 | 66.4 | 61.2 | 7.2 | 20.1 | 48.8 | 18.7 | 2.7 | 10.9 | 40.8 | 7.9 |
| CDS-CZSL* [28] (*CVPR'24*) | 32.1 | 48.0 | 64.7 | 60.4 | - | - | - | - | 2.6 | 10.9 | 38.2 | 8.0 |
| PLID [3] (*ECCV'24*) | 30.8 | 46.6 | 67.6 | 55.5 | 7.3 | 20.4 | 49.1 | 18.7 | 2.5 | 10.6 | 39.1 | 7.5 |
| IMAX [17] (*TPAMI'25*) | 32.3 | 47.5 | 68.4 | 57.3 | 7.6 | 21.4 | 50.2 | 18.6 | 2.6 | 11.2 | 38.7 | 7.9 |
| ClusPro [47] (*ICLR'25*) | 39.5 | 54.1 | 71.0 | **66.2** | **9.3** | **23.0** | **51.2** | **22.1** | 3.0 | 11.6 | 41.6 | 8.3 |
| **TOMCAT (Ours)** | **43.7** | **57.9** | **74.1** | 65.8 | 8.2 | 21.7 | 49.2 | 21.0 | **4.2** | **14.2** | **45.1** | **10.6** |

Table 2: Closed-world and open-world results on Clothing16K.

| Dataset | Methods | Closed-World | | | | Open-World | | | |
|---|---|---|---|---|---|---|---|---|---|
| | | AUC | HM | Seen | Unseen | AUC | HM | Seen | Unseen |
| Clothing16K | SymNet [26] (*CVPR'20*) | 78.8 | 79.3 | 98.0 | 85.1 | 57.4 | 68.3 | 98.2 | 60.7 |
| | CompCos [35] (*CVPR'21*) | 90.3 | 87.2 | 98.5 | 96.8 | 64.1 | 70.8 | 98.2 | 69.8 |
| | CGE [40] (*CVPR'21*) | 89.2 | 84.2 | 98.0 | 97.4 | 62.0 | 68.3 | 98.5 | 69.7 |
| | Co-CGE [36] (*TPAMI'22*) | 88.3 | 87.9 | 98.5 | 94.7 | 59.3 | 69.2 | 98.7 | 63.8 |
| | SCEN [25] (*CVPR'22*) | 78.8 | 78.5 | 98.0 | 89.6 | 53.7 | 61.5 | 96.7 | 62.3 |
| | INV [68] (*ECCV'22*) | 90.6 | 86.6 | 99.0 | 97.0 | 63.6 | 72.0 | 98.7 | 69.0 |
| | OADis [51] (*CVPR'22*) | 88.4 | 86.1 | 97.7 | 94.2 | 53.4 | 63.2 | 98.0 | 58.6 |
| | ADE [13] (*CVPR'23*) | 92.4 | 88.7 | 98.2 | 97.7 | 68.0 | 74.2 | 99.0 | 73.1 |
| | CLPS [18] (*TMM'25*) | 96.2 | 91.7 | 99.2 | 98.9 | 71.5 | 76.1 | 99.2 | 76.1 |
| | ATIF [38] (*TMM'25*) | 96.6 | 95.4 | 99.0 | 99.4 | 91.8 | 92.0 | 99.2 | 93.9 |
| | AIF [38] (*TMM'25*) | 98.2 | 96.3 | 99.5 | 99.6 | 89.9 | 90.3 | 99.7 | 92.2 |
| | **TOMCAT (Ours)** | **99.5** | **98.4** | **99.9** | **100** | **95.8** | **95.0** | **100** | **96.4** |

GIPCOL [59], Troika [14], CDS-CZSL [28], PLID [3], IMAX [17], and ClusPro [47]. TOMCAT is also compared with some canonical methods that conduct their experiments on Clothing16K [68], including SymNet [26], CompCos [35], CGE [40], Co-CGE [36], SCEN [25], INV [68], OADis [51], ADE [13], CLPS [18], ATIF [38], and AIF [38].

## 4.2 Main Results

**Closed-world results** are presented in Table 1 and Table 2. For two fine-grained fashion datasets, our TOMCAT outperforms previous state-of-the-art methods with the improvement of AUC by 1.7%, 1.3%, and HM by 1.7%, 2.1% respectively on UT-Zappos and Clothing16K. Our model also surpasses baselines by 1.1% in AUC and 1.2% in HM on C-GQA. However, TOMCAT falls behind ClusPro and achieves second-best performance on MIT-States, which may be attributed to the substantial

Table 3: Abaltion study of our proposed modules on UT-Zappos and MIT-States. Queue means visual priority queue. T- and V- KAM denote textual and visual KAM. AUW is adaptive update weights.

| Module | | | | UT-Zappos | | | | MIT-States | | | |
|---|---|---|---|---|---|---|---|---|---|---|---|
| Queue | T-KAM | V-KAM | AUW | AUC | HM | Seen | Unseen | AUC | HM | Seen | Unseen |
| | | | | 43.57 | 55.54 | 68.72 | **74.30** | 22.12 | 38.97 | 49.50 | 52.61 |
| ✓ | | | | 43.28 | 55.54 | 68.43 | 74.14 | 22.12 | 39.15 | 49.45 | 52.88 |
| | | ✓ | | 45.68 | 58.22 | 74.39 | 69.01 | 22.18 | 39.22 | 49.50 | **52.95** |
| ✓ | | ✓ | | 43.28 | 55.54 | 68.43 | 74.14 | 22.32 | 39.32 | 49.54 | 52.93 |
| ✓ | ✓ | ✓ | | 46.64 | 58.83 | 76.44 | 68.38 | 22.34 | 39.39 | 49.58 | 52.93 |
| ✓ | ✓ | ✓ | ✓ | **48.31** | **60.18** | **74.99** | 72.77 | **22.55** | **39.45** | **50.32** | **52.95** |

Table 4: Ablation study of our designed loss on UT-Zappos and MIT-States.

| $\mathcal{L}oss$ | | UT-Zappos | | MIT-States | |
|---|---|---|---|---|---|
| $\mathcal{L}_{PE}$ | $\mathcal{L}_{MCRL}$ | AUC | HM | AUC | HM |
| | | 43.57 | 55.54 | 22.12 | 38.97 |
| ✓ | | 44.59 | 57.29 | 22.35 | 39.32 |
| | ✓ | 42.46 | 53.97 | 22.29 | 39.42 |
| ✓ | ✓ | **48.31** | **60.18** | **22.55** | **39.45** |

Table 5: Influence of initialization strategies of KAMs on UT-Zappos and MIT-States.

| Initialization | UT-Zappos | | MIT-States | |
|---|---|---|---|---|
| | AUC | HM | AUC | HM |
| Uniform Random | 43.49 | 56.23 | 21.81 | 38.59 |
| Normal Random | 45.50 | 57.98 | 21.32 | 38.36 |
| Random Walking | 47.08 | 59.12 | 22.14 | 39.16 |
| All Zeros | **48.31** | **60.18** | **22.55** | **39.45** |

Table 6: Influence of test order on UT-Zappos, MIT-States, and C-GQA.

| Test Order | UT-Zappos | | | | MIT-States | | | | C-GQA | | | |
|---|---|---|---|---|---|---|---|---|---|---|---|---|
| | AUC | HM | Seen | Unseen | AUC | HM | Seen | Unseen | AUC | HM | Seen | Unseen |
| Order 1 | 48.31 | 60.18 | 74.49 | 72.77 | 22.55 | 39.45 | 50.32 | 52.95 | 15.98 | 33.95 | 45.30 | 40.12 |
| Order 2 | 48.52 | 60.82 | 74.64 | 72.95 | 22.46 | 39.33 | 49.84 | 53.05 | 15.71 | 33.75 | 45.02 | 39.95 |
| Order 3 | 46.44 | 58.72 | 72.63 | 72.03 | 22.31 | 39.02 | 49.62 | 53.09 | 15.86 | 34.17 | 45.42 | 39.35 |

noise present in the dataset, leading to inaccurate label supervision and hindering the learning of discriminative features.

**Open-world results** are shown in Table 1 and Table 2. Our method performs much better than the second-best method by 4.2%, 1.2%, 4.0% in AUC, and 3.8%, 2.6%, 3.0% in HM on UT-Zappos, C-GQA, and Clothing16K, respectively, which indicates that TOMCAT has great potential in the open-world setting.

**Discussion.** TOMCAT achieves state-of-the-art performance in both closed-world and open-world settings, demonstrating that our proposed method enables the model to adapt to the distribution shift of label space caused by unseen compositions during testing. Particularly in the open-world setting, TOMCAT shows a significant improvement. This can be attributed to the fact that all test images are derived from feasible compositions (both seen and unseen ones), which reinforces our model's focus on the feasible compositions and reduces its attention to infeasible ones.

### 4.3 Ablation Study

In this section, we conduct extensive ablation studies to evaluate the contribution of each component within our proposed TOMCAT on UT-Zappos, MIT-States, and C-GQA in the closed-world setting.

**Ablation Study of Main Modules.** According to Table 3, for MIT-States, empirical results indicate that all proposed modules contribute to the performance improvement of TOMCAT, including the priority queue, multimodal KAMs, and the adaptive update weights. However, on UT-Zappos, incorporating the priority queue and the subsequent visual KAM leads to performance degradation compared to the base model. This phenomenon is attributed to the uniform style of shoes and the subtle inter-composition difference in this dataset, which makes the model insufficient to capture fine-grained semantic distinctions. In contrast, when textual KAM is added, it enables targeted refinement of the textual prototypes based on the stored images. Ultimately, the best performance is

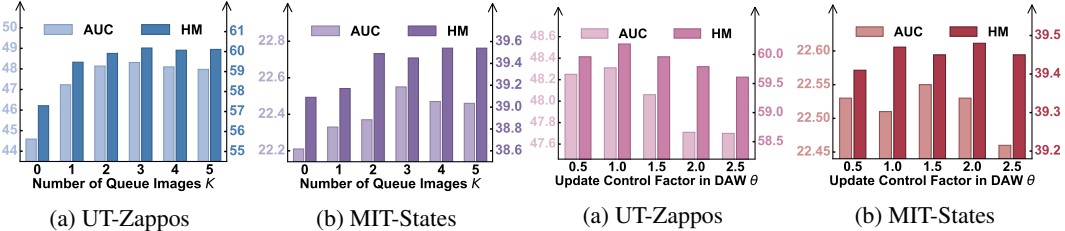

|                          |                          |
| :----------------------: | :----------------------: |
| (a) UT-Zappos            | (b) MIT-States           |

|                          |                          |
| :----------------------: | :----------------------: |
| (a) UT-Zappos            | (b) MIT-States           |

Figure 3: Influence of the number of images in each priority queue $K$ on two datasets.

Figure 4: Influence of the value of update control factor $\theta$ on two datasets.

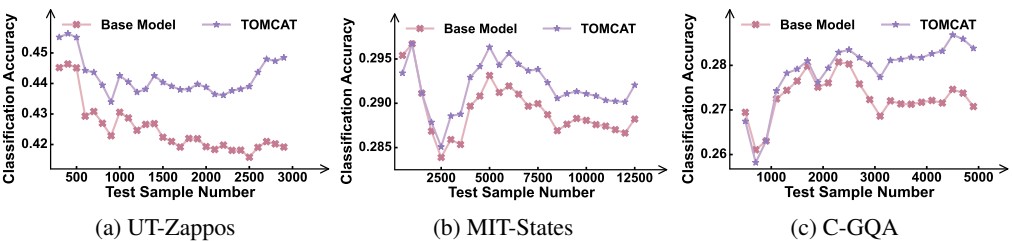

|                    |                    |                    |
| :----------------: | :----------------: | :----------------: |
| (a) UT-Zappos      | (b) MIT-States     | (c) C-GQA          |

Figure 5: Trend of top-1 classification accuracy with increasing test sample size on three datasets.

achieved by TOMCAT with all modules, which indicates that all components are beneficial and that the multimodal KAMs and priority queue offer complementary advantages.

**Ablation Study of Each Loss.** As shown in Table 4, the improvement of adding $\mathcal{L}_{PE}$ to the base model indicates that prediction entropy loss helps the model to adapt towards the new label distribution of seen and unseen compositions. However, using only the constraint term $\mathcal{L}_{MCRL}$ may cause the model to blindly reduce the distance between multimodal prototypes without considering other factors, thereby degrading the model's performance. Finally, by combining $\mathcal{L}_{PE}$ and $\mathcal{L}_{MCRL}$, TOMCAT achieves the best performance, which confirms the effectiveness of our algorithmic designs.

**Influence of Different Initialization Strategies of Multimodal KAMs.** Table 5 shows that the zero-initialization outperforms other random initialization strategies. The reason is that starting with random initialization at test-time causes the model to forget the knowledge gained during training.

**Influence of Test Order.** Since TOMCAT continually accumulates knowledge during testing, the test order may affect the results. We conduct three experiments with different random seeds to vary the sample order. The results in Table 6 suggest that there exists performance variance among different orders, although the differences are not statistically significant.

**Influence of Hyperparameters.** In Fig. 3 and Fig. 4, we explore the influence of the number of images stored in priority queue $K$ and the update control factor $\theta$ respectively on UT-Zappos and MIT-States. TOMCAT achieves the best performance when $K = 3$ on both datasets, as a small number of images causes instability in the visual prototypes, while a large number may include low-confidence images. Moreover, as $\theta$ increases, the performance initially improves but subsequently deteriorates. Analysis reveals a small $\theta$ leads to insufficient adaptation of the prototypes when encountering images with substantial variation, whereas a large one results in over-updating of the prototypes.

### 4.4 Qualitative Analysis

In this section, we conduct a qualitative analysis of TOMCAT to visualize its predictions on UT-Zappos, MIT-States, and C-GQA in the closed-world setting.

**Trend of Classification Accuracy.** To verify whether TOMCAT continuously improves, we visualize in Fig. 5 how top-1 classification accuracy evolves with the number of test samples on three datasets. At the beginning, TOMCAT performs comparably to the base model; however, as more test samples arrive, TOMCAT progressively demonstrates its superiority. The performance gap suggests TOMCAT continuously accumulates useful knowledge to overcome the distribution shift of the label space.

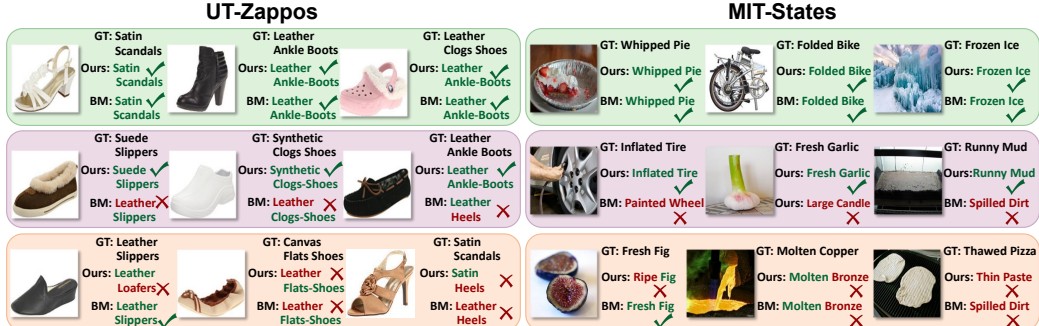

Figure 6: Case study on UT-Zappos and MIT-States. We compare TOMCAT (Ours) with the base model (BM) after training. The successful and failure results are marked in green and red, respectively.

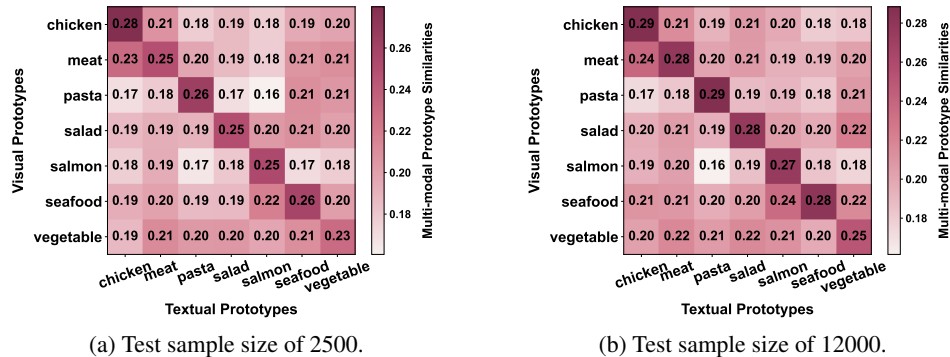

(a) Test sample size of 2500.        (b) Test sample size of 12000.

Figure 7: Similarity heatmap of multimodal prototypes on MIT-States. All unseen compositions consisting of the attribute `cooked` and its corresponding objects (*e.g.*, `chicken`, `meat`...) are selected.

**Successful and Failure Cases.** In Fig. 6, we report top-1 predictions for some image cases. It can be observed that benefiting from adapting to the new distribution, TOMCAT shows superior performance with higher accuracy. For the failure cases, the predictions of TOMCAT can also describe them; for instance, the image of `satin sandals` can also be categorized as `satin heels`.

**Similarity Heatmap of Multimodal Prototypes.** We visualize the cosine similarities between multimodal prototypes of all unseen compositions consisting of the attribute `cooked` and its corresponding objects in Fig. 7. Compared to the test size of 2500, the diagonal elements become larger at the test size of 12000, indicating that TOMCAT enhances the connection between multimodal information. Meanwhile, the increase in off-diagonal elements is attributed to all used compositions involving `cooked` food (almost in the same cluster), where larger similarities can better characterize their semantic structure.

## 5  Conclusion

In this work, we consider the issue of label distribution shift in CZSL, which arises from novel compositions recomposed from attributes and objects. Therefore, we propose TOMCAT to accumulate comprehensive knowledge of visual and textual modalities from unsupervised data at test time to overcome the challenge. Specifically, we update textual and visual prototypes by multimodal knowledge accumulation modules, and use adaptive update weights to control the degree. Meanwhile, a priority queue is leveraged that stores historical images to obtain the visual knowledge. We hope that our work will inspire further research into exploring label space shift at test time in CZSL.

**Limitation.** While our proposed TOMCAT accumulates knowledge from test samples to overcome distribution shift of label space at test time, two potential limitations are identified: (1) TOMCAT accumulates comprehensive knowledge only at the level of compositions, neglecting to exploit the information contained within each primitive. (2) Without proper initialization of the priority queue, the model is prone to bias toward the compositions of images already stored during the testing phase.

## Acknowledgements

This work was supported by the Fundamental Research Funds for the Central Universities (2025YJS052 and 2025JBZX059), the Beijing Natural Science Foundation (4242046), and the Hebei Province Natural Science Foundation (F2025105018).

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

# Appendix

## A   Visual Adapter

Based on AdapterFormer [5], a set of learnable adapters is injected in parallel with the multi-head attention layer and feed-forward layer, respectively within each Transformer block of the CLIP visual encoder, while keeping the initial parameters frozen, as shown in Fig. 8. Given the input embedding $e$ of each layer, the adapter is formulated as:

$$Adapter(e) = \mathbf{W}^{\text{up}}(ReLU(\mathbf{W}^{\text{down}}e)) \,, \tag{10}$$

where $\mathbf{W}^{\text{down}}$ and $\mathbf{W}^{\text{up}}$ are learnable linear layers employed for down-sampling and up-sampling, respectively. For each layer, the final output is obtained by adding the adapter output to the original layer output through a residual connection.

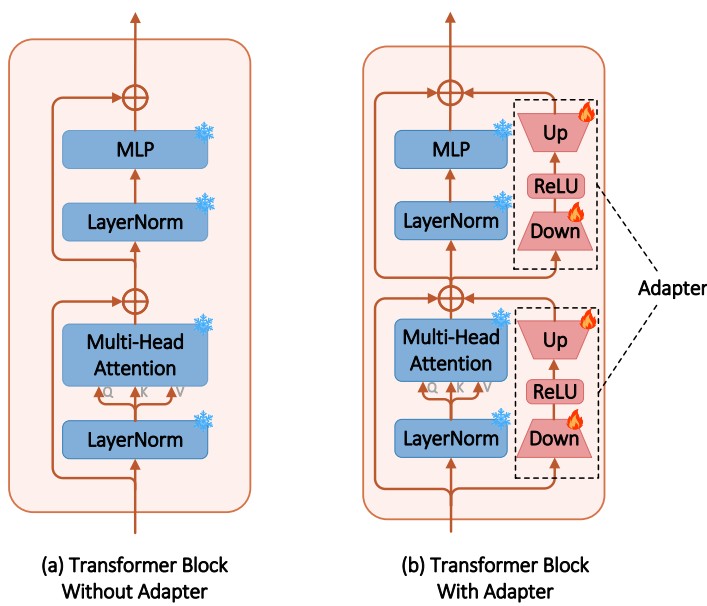

Figure 8: Transformer block within visual encoder of CLIP without and with Adapter.

## B   Theoretical Explanation of Prediction Entropy

In the absence of labeled target-composition samples, the Minimum Entropy Principle is adopted to improve model predictions. This principle is grounded in the cluster assumption of semi-supervised learning, which posits that the decision boundary should pass through low-density regions in the prediction space. By minimizing the entropy of the model predictions, we encourage high-confidence outputs and implicitly shift the decision boundary away from high-density data regions, thus reducing classification ambiguity.

With reference to [12], we formally define $x \sim \mathcal{D}_T$ as an unlabeled input, and define $p(y|x)$ as the predicted class distribution from the model. The prediction entropy of a single example is defined as:

$$\mathcal{H}(p(y|x)) = -\sum_{c=1}^{N_C} p(y = c|x) \log p(y = c|x) \,, \tag{11}$$

where $N_C$ is the total number of classes. The entropy minimization objective is expressed as:

$$\mathcal{L}_{entropy} = \mathbb{E}_{x \sim \mathcal{D}_T} \left[ \mathcal{H}(p(y|x)) \right] \,. \tag{12}$$

Assuming the target data follows a continuous density distribution $q(x)$, the entropy loss becomes:

$$\mathcal{L}_{entropy} = \int \mathcal{H}(p(y|x))q(x) \, dx \,. \tag{13}$$

This formulation reveals that high-density regions (*i.e.*, large $q(x)$) contribute more to the loss. Therefore, minimizing this loss leads the model to produce low-entropy (*i.e.*, high-confidence) predictions in those regions, which implicitly encourages the decision boundary to lie in low-density areas of the prediction space.

On the other hand, to understand how entropy minimization sharpens model predictions, we analyze the gradient behavior in the prediction space. Let the model output logits $z = (z_1, \ldots, z_{N_C})$, and the predicted probabilities be given by the softmax function:

$$p_i = \frac{e^{z_i}}{\sum_{j=1}^{N_C} e^{z_j}}. \tag{14}$$

Based on this, the gradient of entropy is computed with respect to the logits $z_i$ using the chain rule. The derivative of $\mathcal{H}(p)$ with respect to $p_i$ is:

$$\frac{\partial \mathcal{H}}{\partial p_i} = -\log p_i - 1, \tag{15}$$

and the Jacobian of the softmax function is:

$$\frac{\partial p_i}{\partial z_j} = \begin{cases} p_i(1 - p_i), & \text{if } i = j, \\ -p_i p_j, & \text{if } i \neq j. \end{cases} \tag{16}$$

Therefore, the gradient of entropy with respect to $z_i$ becomes:

$$\frac{\partial \mathcal{H}}{\partial z_i} = \sum_{j=1}^{N_C} \frac{\partial \mathcal{H}}{\partial p_j} \cdot \frac{\partial p_j}{\partial z_i}. \tag{17}$$

This gradient encourages the logits to diverge, making one class score dominant over others, and thus increasing the confidence of the softmax prediction (*i.e.*, moving $p$ closer to a one-hot vector). As a result, entropy minimization leads to sharper decision boundaries and effectively separates data clusters in the prediction space.

This microscopic gradient behavior aligns with the macroscopic Minimum Entropy Principle, both of which serve to place the decision boundary in low-density regions and enforce prediction consistency within high-density clusters.

## C Dataset Introduction

We conduct our experiments on four CZSL datasets, *i.e.*, UT-Zappos [61], MIT-States [15], C-GQA [40], and Clothing16K [68]. UT-Zappos is a fine-grained fashion dataset consisting of different shoes (*e.g.*, heels, sandals) with their material (*e.g.*, leather, satin). MIT-States contains diverse real-world objects (*e.g.*, envelope, tower) and attributes (*e.g.*, folded, tight) collected by early image search engine technology. However, it suffers from considerable noise, with approximately 70% of the labels being incorrect [2, 68, 60]. C-GQA is a large-scale dataset composed of in-the-wild images with more general compositions (*e.g.*, marble countertop, striped pillow). For the low-noise dataset Clothing16K, the objects are mainly various kinds of clothing (*e.g.*, dress, shirt), while the attributes are the clothing colors (*e.g.*, pink, white). Table 7 reports the common data splits of the four datasets.

Table 7: Summary statistics of the four datasets used in our experiments.

| Dataset | Composition | | | Train | | Validation | | | Test | | |
|---|---|---|---|---|---|---|---|---|---|---|---|
| | $|A|$ | $|O|$ | $|A \times O|$ | $|C^s|$ | $|\mathcal{X}|$ | $|C^s|$ | $|C^u|$ | $|\mathcal{X}|$ | $|C^s|$ | $|C^u|$ | $|\mathcal{X}|$ |
| UT-Zappos [61] | 16 | 12 | 192 | 83 | 22998 | 15 | 15 | 3214 | 18 | 18 | 2914 |
| MIT-States [15] | 115 | 245 | 28175 | 1262 | 30338 | 300 | 300 | 10420 | 400 | 400 | 12995 |
| C-GQA [40] | 413 | 674 | 278362 | 5592 | 26920 | 1252 | 1040 | 7280 | 888 | 923 | 5098 |
| Clothing16K [68] | 9 | 8 | 72 | 18 | 7242 | 10 | 10 | 5515 | 9 | 8 | 3413 |

## D  Implementation Details

We use a single NVIDIA RTX 3090 GPU to train the base model during training and TOMCAT at test time using mixed-precision training [37] under the PyTorch framework [45]. The trainable prompt of CLIP ViT-L/14 in the training phase is initialized by "*a photo of*". The hyperparameters for the four datasets are listed in Table 8.

Table 8: Hyperparameter settings for UT-Zappos, MIT-States, C-GQA, and Clothing16K.

| Hyperparameters | UT-Zappos | MIT-States | C-GQA | Clothing16K |
|---|---|---|---|---|
| **The Base Model (*Training Phase*)** | | | | |
| Batch Size | 128 | 64 | 16 | 128 |
| Epochs | 20 | 20 | 20 | 20 |
| Prompt Dropout Rate | 0.3 | 0.3 | 0 | 0.3 |
| Adapter Downsampling Dimension | 64 | 64 | 64 | 64 |
| Adapter Dropout | 0.1 | 0.1 | 0.1 | 0.1 |
| Optimizer | Adam | Adam | Adam | Adam |
| Optimizer-Weight Decay | 1e-5 | 1e-4 | 1e-5 | 1e-5 |
| Optimizer-Learning Rate | 5e-4 | 1e-4 | 1e-4 | 5e-4 |
| Scheduler | StepLR | StepLR | StepLR | StepLR |
| Scheduler-Step Size | 5 | 5 | 5 | 5 |
| Scheduler-Gamma | 0.5 | 0.5 | 0.5 | 0.5 |
| **TOMCAT (*Test Phase*)** | | | | |
| Batch Size | 1 | 1 | 1 | 1 |
| Image Number of Priority Queue | 3 | 3 | 3 | 3 |
| Optimizer | AdamW | AdamW | AdamW | AdamW |
| Optimizer-Epsilon | 1e-3 | 1e-3 | 1e-3 | 1e-3 |
| Optimizer-Weight Decay | 1e-3 | 1e-4 | 1e-4 | 1e-3 |
| Optimizer-Learning Rate | 5e-6 | 1e-6 | 6.25e-6 | 5e-6 |
| $\alpha$ | 0 | 1.25 | 0.5 | 0.25 |
| $\beta$ | - | 10 | 10 | 7.5 |
| $\theta$ | 1 | 1.5 | 2 | 1 |
| $\lambda$ | 3.5 | 2.5 | 1.75 | 3.5 |

## E  Comparison with More Baselines

In this section, we compare our TOMCAT with three types of baselines on UT-Zappos, MIT-States, and C-GQA: (1) more CZSL methods; (2) online test-time adaptation (OTTA) of VLM methods; and (3) multimodal large language models (MLLMs).

**Comparison with More CZSL Methods.** TOMCAT is compared with more state-of-the-art CZSL methods on UT-Zappos [61], MIT-States [15], and C-GQA [40] in both closed-world and open-world settings in Table 17 and Table 18, respectively. Relative to Table 1, these additional baselines include LE+ [39], AttOp [41], TMN [46], SymNet [26], SCEN [25], OADis [51], ADE [13], CANET [56], CLPS [18], DBC [67], CoP [69], VisProd [20], KG-SP [21], and SAD-SP [33]. Our TOMCAT surpasses all other models and achieves state-of-the-art performance.

**Comparison with OTTA of VLM Methods.** Building on the base model obtained after training, we conduct experiments on the three datasets with OTTA of VLM approaches, including online-TPS [53], TDA [19], and DPE [63]. These methods also utilize unlabeled test data, making the comparison fairer, although the original comparison with CZSL methods is fair due to the absence of labels of test samples. In Table 9, TOMCAT yields 3.7%, 0.4%, and 0.4% AUC gains compared with OTTA of VLM methods on the three datasets, which demonstrates that TOMCAT also exhibits superior performance over OTTA methods on CZSL task under fair conditions.

**Comparison with MLLMs.** With the advancement of MLLMs, they can also be employed to generate image descriptions or categories. Therefore, we compare TOMCAT with two outstanding MLLMs (*i.e.*, LLaVA v1.5 [31] and InternVL-3 [75]) on UT-Zappos and MIT-States. Specifically, we provide the prompt "*Classify the image and output strictly two words in the form: 'attribute object' (an attribute must be generated). No other text. Example: 'red apple'. Choose the attribute and object from their respective list. Attribute list:* [Attribute List]. *Object list:* [Object List]*.*" to

Table 9: Comparison with OTTA of VLM methods in the closed-world setting on UT-Zappos, MIT-States, and C-GQA.

| Closed-world Methods | UT-Zappos | | | | MIT-States | | | | C-GQA | | | |
|---|---|---|---|---|---|---|---|---|---|---|---|---|
| | AUC | HM | Seen | Unseen | AUC | HM | Seen | Unseen | AUC | HM | Seen | Unseen |
| online-TPS [53] | 44.6 | 57.3 | 71.3 | 71.4 | 22.2 | 39.1 | 49.6 | 52.8 | 15.6 | 33.8 | 45.0 | 39.6 |
| TDA [19] | 41.6 | 54.6 | 69.0 | 70.6 | 22.0 | 38.7 | 49.5 | 52.9 | 14.4 | 32.0 | 44.7 | 37.2 |
| DPE [63] | 43.3 | 56.0 | 68.5 | **74.2** | 22.0 | 38.9 | 49.2 | 52.8 | 15.3 | 33.5 | 44.0 | 39.7 |
| **TOMCAT (Ours)** | **48.3** | **60.2** | **74.5** | 72.8 | **22.6** | **39.5** | **50.3** | **53.0** | **16.0** | **34.0** | **45.3** | **40.1** |

Table 10: Comparison with MLLMs on UT-Zappos and MIT-States.

| Methods | UT-Zappos | | | | MIT-States | | | |
|---|---|---|---|---|---|---|---|---|
| | Comp. ↑ | Attr.↑ | Obj. ↑ | Time (ms) ↓ | Comp. ↑ | Attr. ↑ | Obj. ↑ | Time (ms) ↓ |
| LLaVA v1.5 [31] | 0.32 | 10.17 | 2.13 | 463 | 5.72 | 1.06 | 21.77 | 857 |
| InternVL-3 [75] | 2.06 | 8.77 | 23.71 | 5770 | 4.43 | 8.53 | 48.28 | 728 |
| **TOMCAT (Ours)** | **44.88** | **58.54** | **77.45** | **47** | **30.92** | **41.00** | **56.58** | **145** |

MLLMs and obtain their predictions. Since MLLMs cannot produce class probabilities for computing AUC, we report the top-1 classification accuracy (%) for compositions (**Comp.**), attributes (**Attr.**), objects (**Obj.**), and the inference **Time** in Table 10. TOMCAT significantly outperforms them with less time overhead, indicating that MLLMs are primarily designed for text generation and are less effective for representation learning tasks (*e.g.*, recognition and retrieval). Moreover, the behavior of generative models is inherently difficult to control, making them unsuitable for recognition tasks.

# F   More Ablation Study

In this section, more ablation studies are presented in the closed-world setting.

**Ablation study on fine-tuning CLIP with the training set.**   In Table 11, we respectively use the naive CLIP without fine-tuning on the training set and the base model fine-tuned on the training data as the starting points for testing. The latter significantly outperforms the former, indicating that while the naive CLIP exhibits strong zero-shot abilities, its general-purpose nature limits its effectiveness in the specific downstream task requiring composition reasoning under the zero-shot setting.

**Influence of learning rate.**   Since our method simulates test-time improvement in real-world applications, the value of learning rate is crucial. According to Table 12, TOMCAT achieves optimal performance when the learning rate is set to 5e-6.

Table 11: Ablation study of fine-tuning CLIP on the training set of three datasets, respectively.

| Dataset | tuning | AUC | HM | Seen | Unseen |
|---|---|---|---|---|---|
| UT-Zappos | ✗ | 1.54 | 7.29 | 4.79 | 50.66 |
| | ✓ | **48.31** | **60.18** | **74.49** | **72.77** |
| MIT-States | ✗ | 11.42 | 26.79 | 30.88 | 46.50 |
| | ✓ | **22.55** | **39.45** | **50.32** | **52.95** |
| C-GQA | ✗ | 1.47 | 8.84 | 7.79 | 25.09 |
| | ✓ | **15.98** | **33.95** | **45.30** | **40.12** |

Table 12: Influence of learning rate (lr) of TOMCAT on UT-Zappos.

| lr | AUC | HM | Seen | Unseen |
|---|---|---|---|---|
| 1e-3 | 17.11 | 17.11 | 54.25 | 39.77 |
| 1e-4 | 28.66 | 46.14 | 70.19 | 45.21 |
| 1e-5 | 47.28 | 58.96 | **76.25** | 69.65 |
| 5e-6 | **48.31** | **60.18** | 74.49 | 72.77 |
| 1e-6 | 45.78 | 57.65 | 70.97 | 74.25 |
| 1e-7 | 43.69 | 55.77 | 68.82 | **74.35** |

**Influence of the hyperparameter controlling multimodal balance** $\alpha$.   In table 13, TOMCAT achieves the best performance on MIT-States when $\alpha$ is set to 1.25. An excessively large or small value of $\alpha$ leads the model to overemphasize one modality while neglecting the other, thereby undermining its discriminative capability.

**Influence of the weighting coefficient of** $\mathcal{L}_{MCRL}$ $\lambda$.   In TOMCAT, balancing the contributions of each loss term plays a crucial role. As shown in Table 14, the best performance on MIT-States is

achieved when $\lambda = 1.5$, indicating that an appropriate trade-off between the competing objectives leads to optimal model behavior.

Table 13: Influence of $\alpha$ on MIT-States.

| $\alpha$ | AUC | HM | Seen | Unseen |
|---|---|---|---|---|
| 0.25 | 22.24 | 39.27 | 49.58 | 52.86 |
| 0.5 | 22.20 | 39.22 | 49.50 | 52.81 |
| 0.75 | 22.34 | **39.47** | 49.75 | 52.89 |
| 1 | 22.36 | 39.38 | 49.92 | 52.83 |
| 1.25 | **22.55** | 39.45 | **50.32** | **52.95** |
| 1.5 | 22.43 | 39.31 | 49.96 | **52.95** |
| 1.75 | 22.41 | 39.31 | 49.79 | 53.04 |
| 2 | 22.42 | 39.17 | 49.79 | 53.12 |

Table 14: Influence of $\lambda$ on MIT-States.

| $\lambda$ | AUC | HM | Seen | Unseen |
|---|---|---|---|---|
| 0.25 | 22.50 | 39.46 | 50.08 | **52.99** |
| 0.5 | 22.53 | 39.41 | 50.13 | 52.93 |
| 0.75 | 22.50 | 39.40 | 50.04 | 52.92 |
| 1 | 22.51 | **39.47** | 50.04 | 52.91 |
| 1.25 | 22.48 | 39.42 | 50.00 | 52.93 |
| 1.5 | **22.55** | 39.45 | **50.32** | 52.95 |
| 1.75 | 22.53 | 39.48 | 50.17 | 52.91 |
| 2 | 22.46 | 39.45 | 50.01 | 52.92 |

**Ablation Study of Prediction Probability in Eq. 6.** In Table 15, we observe that relying solely on visual prototypes yields poor performance, likely due to the large intra-class variance in images, which limits the discriminative power of the model. Incorporating both textual and visual prototypes better exploits the complementary multimodal strengths, enabling TOMCAT to achieve good performance.

Table 15: Ablation Study of prediction probability on three datasets. Text- and visual-only denote that the prediction probability is derived from only textual prototypes and visual prototypes, respectively.

| Prediction Probability | UT-Zappos | | | | MIT-States | | | | C-GQA | | | |
|---|---|---|---|---|---|---|---|---|---|---|---|---|
| | AUC | HM | Seen | Unseen | AUC | HM | Seen | Unseen | AUC | HM | Seen | Unseen |
| Text-only | 48.31 | 60.18 | 74.49 | 72.77 | 22.1 | 38.95 | 49.33 | 52.8 | 15.79 | 33.76 | 45.03 | 39.86 |
| Visual-only | 1.66 | 0 | 67.45 | 4.92 | 15.66 | 33.76 | 16.76 | 41.48 | 3.95 | 16.65 | 29.29 | 14.95 |
| Multimodal (TOMCAT) | **48.31** | **60.18** | **74.49** | **72.77** | **22.55** | **39.45** | **50.32** | **52.95** | **15.98** | **33.95** | **45.30** | **40.12** |

## G    More Qualitative Analysis

In this section, we report more qualitative analysis of TOMCAT in the closed-world setting.

**Time and memory analysis.** As shown in Table 16, we present the time, latency, and memory occupation of the base model and TOMCAT. Our TOMCAT achieves substantial performance improvement with minimal additional computational resources at test time.

Table 16: Comparison of time, latency, and memory between TOMCAT and the base model (w/o TOMCAT) on UT-Zappos and MIT-States. Time means that average testing time across all samples. Latency indicates time from input to prediction output for per sample (excluding backpropagation). Memory represents the GPU memory occupied by the model during testing.

| Dataset | Time | | Latency | | Memory | |
|---|---|---|---|---|---|---|
| | TOMCAT | w/o TOMCAT | TOMCAT | w/o TOMCAT | TOMCAT | w/o TOMCAT |
| UT-Zappos | 47ms | 36ms | 35ms | 32ms | 4044MB | 3801MB |
| MIT-States | 145ms | 70ms | 112ms | 37ms | 4180MB | 4175MB |

**More case study.** In Fig. 9 and Fig. 10, we present more case studies on UT-Zappos and MIT-States, respectively. As shown, by leveraging our designed algorithm with unlabeled data at test time, TOMCAT significantly improves the accuracy of composition predictions compared to the base model. In some failure cases (*e.g.*, the image labeled as `new bus` in MIT-States), our method is also capable of making correct predictions, though it focuses on aspects different from the labels.

## H    Broader Impacts

In this work, we recognize that the unseen compositions resulting from the recombination of attributes and objects lead to the distribution shift of label space. Therefore, we aim to establish a more reliable

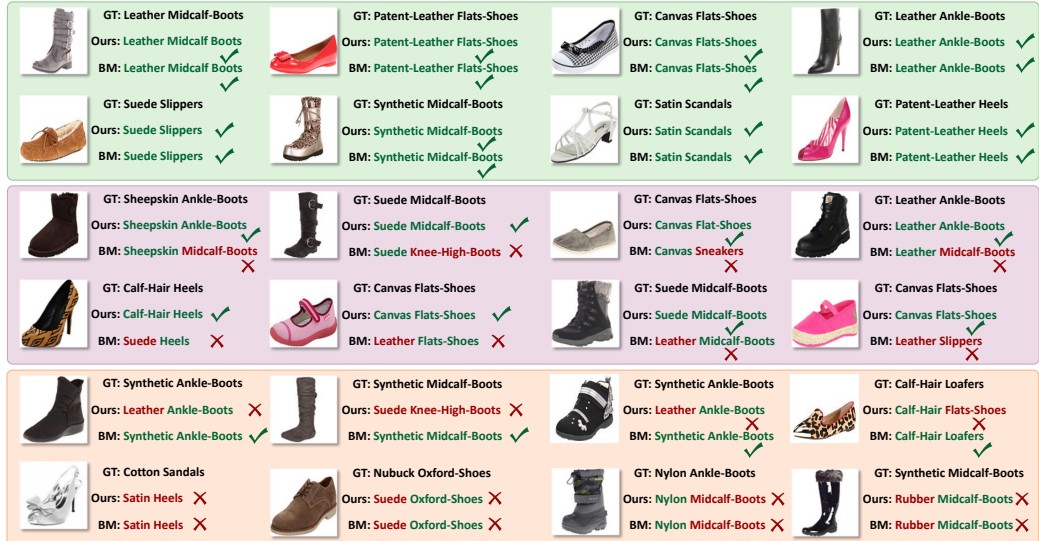

Figure 9: More case study on UT-Zappos. We compare TOMCAT (Ours) with the base model (BM) after training. The successful and failure results are marked in green and red, respectively.

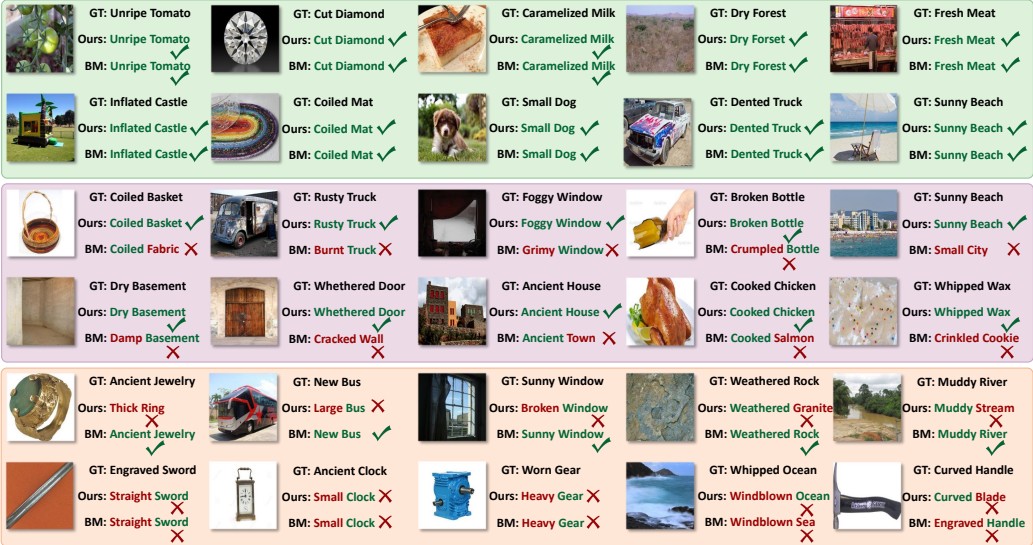

Figure 10: More case study on MIT-States.

compositional reasoning system by leveraging test samples to enhance composition modeling, which is more in line with the application and development in real-world scenarios. To the best of our knowledge, our method has no potential negative societal impact.

## I Licenses

We have explicitly cited the datasets and models used in our paper. Their licenses are listed as follows.

**Datasets.** UT-Zappos [61] and MIT-States [15] are two early-proposed datasets commonly used in CZSL. Their creators and owners have not declared any license and have allowed non-commercial research use. C-GQA [40] is under CC BY 4.0 license. Clothing16K [68] is under CC0 license.

**Models.** In this work, our TOMCAT is built on existing code repositories: CLIP [49], CoOp [73], AdapterFormer [5], Troika [14], TDA [19], TPS [53], and DPE [63]. The code implementations used in our paper are all under MIT license.

Table 17: Comparison with more baselines in the closed-world setting on UT-Zappos, MIT-States, and C-GQA. The best results are displayed in **boldface**, and the second-best results are underlined.

| Closed-world Methods | UT-Zappos | | | | MIT-States | | | | C-GQA | | | |
|---|---|---|---|---|---|---|---|---|---|---|---|---|
| | AUC | HM | Seen | Unseen | AUC | HM | Seen | Unseen | AUC | HM | Seen | Unseen |
| LE+ [39] (CVPR'17) | 25.7 | 41.0 | 53.0 | 61.9 | 2.0 | 10.7 | 15.0 | 20.1 | 0.6 | 5.3 | 16.1 | 5.0 |
| AttOp [41] (CVPR'18) | 25.9 | 40.8 | 59.8 | 54.2 | 1.6 | 9.9 | 14.3 | 17.4 | 0.3 | 2.9 | 11.8 | 3.9 |
| TMN [46] (ICCV'19) | 29.3 | 45.0 | 58.7 | 60.0 | 2.9 | 13.0 | 20.2 | 20.1 | 1.1 | 7.7 | 21.6 | 6.3 |
| SymNet [26] (CVPR'20) | 23.9 | 39.2 | 53.3 | 57.9 | 3.0 | 16.1 | 24.4 | 25.2 | 1.8 | 9.8 | 25.2 | 9.2 |
| SCEN [25] (CVPR'22) | 32.0 | 47.8 | 63.5 | 63.1 | 5.3 | 18.4 | 29.9 | 25.2 | 2.9 | 12.4 | 28.9 | 12.1 |
| OADis [51] (CVPR'22) | 30.0 | 44.4 | 59.5 | 65.5 | 5.9 | 18.9 | 31.1 | 25.6 | - | - | - | - |
| ADE [13] (CVPR'23) | 35.1 | 51.1 | 63.0 | 64.3 | - | - | - | - | 5.2 | 18.0 | 35.0 | 17.7 |
| CANET [56] (CVPR'23) | 33.1 | 47.3 | 61.0 | 66.3 | 5.4 | 17.9 | 29.0 | 26.2 | 3.3 | 14.5 | 30.0 | 13.2 |
| CLPS [18] (TMM'25) | 37.2 | 51.8 | 63.2 | 70.1 | - | - | - | - | 5.6 | 18.9 | 35.1 | 19.0 |
| DBC [67] (TPAMI'25) | 35.4 | 51.3 | 63.0 | 67.5 | - | - | - | - | 5.0 | 18.0 | 35.2 | 17.8 |
| CLIP [49] (ICML'21) | 5.0 | 15.6 | 15.8 | 49.1 | 11.0 | 26.1 | 30.2 | 46.0 | 1.4 | 8.6 | 7.5 | 25.0 |
| CoOp [73] (IJCV'22) | 18.8 | 34.6 | 52.1 | 49.3 | 13.5 | 29.8 | 34.4 | 47.6 | 4.4 | 17.1 | 20.5 | 26.8 |
| Co-CGE [36] (TPAMI'22) | 36.3 | 49.7 | 63.4 | 71.3 | 17.0 | 33.1 | 46.7 | 45.9 | 5.7 | 18.9 | 34.1 | 21.2 |
| CSP [42] (ICLR'23) | 33.0 | 46.6 | 64.2 | 66.2 | 19.4 | 36.3 | 46.6 | 49.9 | 6.2 | 20.5 | 28.8 | 26.8 |
| DFSP(i2t) [34] (CVPR'23) | 32.1 | 45.1 | 64.2 | 66.4 | 20.7 | 37.2 | 47.4 | 52.4 | 8.7 | 24.3 | 35.6 | 29.3 |
| DFSP(BiF) [34] (CVPR'23) | 33.5 | 47.1 | 63.3 | 69.2 | 20.8 | 37.7 | 47.1 | 52.8 | 9.9 | 26.2 | 36.5 | 32.0 |
| DFSP(t2i) [34] (CVPR'23) | 36.0 | 47.2 | 66.7 | 71.7 | 20.6 | 37.3 | 46.9 | 52.0 | 10.5 | 27.1 | 38.2 | 32.0 |
| GIPCOL [59] (WACV'24) | 36.2 | 48.8 | 65.0 | 68.5 | 19.9 | 36.6 | 48.5 | 49.6 | 7.1 | 22.5 | 31.9 | 28.4 |
| Troika [14] (CVPR'24) | 41.7 | 54.6 | 66.8 | 73.8 | 22.1 | 39.3 | 49.0 | 53.0 | 12.4 | 29.4 | 41.0 | 35.7 |
| CDS-CZSL [28] (CVPR'24) | 39.5 | 52.7 | 63.9 | 74.8 | 22.4 | 39.2 | 50.3 | 52.9 | 11.1 | 28.1 | 38.3 | 34.2 |
| CoP [69] (ICME'24) | 36.2 | 51.2 | 64.8 | 67.3 | 19.7 | 36.4 | 47.0 | 50.9 | 7.3 | 21.7 | 33.3 | 27.0 |
| PLID [3] (ECCV'24) | 38.7 | 52.4 | 67.3 | 68.8 | 22.1 | 39.0 | 49.7 | 52.4 | 11.0 | 27.9 | 38.8 | 33.0 |
| IMAX [17] (TPAMI'25) | 40.6 | 54.2 | 69.3 | 70.7 | 21.9 | 39.1 | 48.7 | 53.8 | 12.8 | 29.8 | 39.7 | 35.8 |
| ClusPro [47] (ICLR'25) | 46.6 | 58.5 | 70.7 | 76.0 | 23.8 | 40.7 | 52.1 | 54.0 | 14.9 | 32.8 | 44.3 | 37.8 |
| **TOMCAT (Ours)** | 48.3 | 60.2 | 74.5 | 72.8 | 22.6 | 39.5 | 50.3 | 53.0 | 16.0 | 34.0 | 45.3 | 40.1 |

Table 18: Comparison with more baselines in the open-world setting on UT-Zappos, MIT-States, and C-GQA. We report the results of CDS-CZSL* [28] using the same post-training feasibility calibration [35] as our TOMCAT and other baselines use.

| Open-world Methods | UT-Zappos | | | | MIT-States | | | | C-GQA | | | |
|---|---|---|---|---|---|---|---|---|---|---|---|---|
| | AUC | HM | Seen | Unseen | AUC | HM | Seen | Unseen | AUC | HM | Seen | Unseen |
| LE+ [39] (CVPR'17) | 16.3 | 30.5 | 60.4 | 36.5 | 0.3 | 2.7 | 14.2 | 2.5 | 0.1 | 1.0 | 19.2 | 0.7 |
| AttOp [41] (CVPR'18) | 13.7 | 29.4 | 50.9 | 34.2 | 0.7 | 4.7 | 16.6 | 5.7 | - | - | - | - |
| TMN [46] (ICCV'19) | 8.4 | 21.7 | 55.9 | 18.1 | 0.1 | 1.2 | 12.6 | 0.9 | - | - | - | - |
| SymNet [26] (CVPR'20) | 18.5 | 34.5 | 53.3 | 44.6 | 0.8 | 5.8 | 21.4 | 7.0 | 0.4 | 3.3 | 26.7 | 2.2 |
| VisProd [20] (NeurIPS'21) | 19.7 | 36.9 | 54.6 | 42.8 | 0.7 | 5.6 | 20.9 | 5.8 | 0.3 | 2.8 | 24.8 | 1.7 |
| KG-SP [21] (CVPR'22) | 26.5 | 42.3 | 61.8 | 52.1 | 1.3 | 7.4 | 28.4 | 7.5 | 0.8 | 4.7 | 31.5 | 2.9 |
| ADE [13] (CVPR'23) | 27.1 | 44.8 | 62.4 | 50.7 | - | - | - | - | 1.4 | 7.6 | 35.1 | 4.8 |
| SAD-SP [33] (TPAMI'24) | 28.4 | 44.0 | 63.1 | 54.7 | 1.4 | 7.8 | 29.1 | 7.6 | 1.0 | 5.9 | 31.0 | 3.9 |
| DBC [67] (TPAMI'25) | 28.6 | 44.9 | 63.0 | 52.8 | - | - | - | - | 1.4 | 7.6 | 35.6 | 4.7 |
| CLIP [49] (ICML'21) | 2.2 | 11.2 | 15.7 | 20.6 | 3.0 | 12.8 | 30.1 | 14.3 | 0.3 | 4.0 | 7.5 | 4.6 |
| CoOp [73] (IJCV'22) | 13.2 | 28.9 | 52.1 | 31.5 | 2.8 | 12.3 | 34.6 | 9.3 | 0.7 | 5.5 | 21.0 | 4.6 |
| Co-CGE [36] (TPAMI'22) | 28.4 | 45.3 | 59.9 | 56.2 | 5.6 | 17.7 | 38.1 | 20.0 | 0.9 | 5.3 | 33.2 | 3.9 |
| CSP [42] (ICLR'23) | 22.7 | 38.9 | 64.1 | 44.1 | 5.7 | 17.4 | 46.3 | 15.7 | 1.2 | 6.9 | 28.7 | 5.2 |
| DFSP(i2t) [34] (CVPR'23) | 26.4 | 41.2 | 64.3 | 53.8 | 6.7 | 19.1 | 47.2 | 18.2 | 2.0 | 9.0 | 35.6 | 6.5 |
| DFSP(BiF) [34] (CVPR'23) | 27.6 | 42.7 | 63.5 | 57.2 | 6.7 | 19.2 | 47.1 | 18.1 | 2.4 | 10.6 | 36.4 | 7.6 |
| DFSP(t2i) [34] (CVPR'23) | 30.3 | 44.0 | 66.8 | 60.0 | 6.8 | 19.3 | 47.5 | 18.5 | 2.4 | 10.4 | 38.3 | 7.2 |
| GIPCOL [59] (WACV'24) | 23.5 | 40.1 | 65.0 | 45.0 | 6.3 | 17.9 | 48.5 | 16.0 | 1.3 | 7.3 | 31.6 | 5.5 |
| Troika [14] (CVPR'24) | 33.0 | 47.8 | 66.4 | 61.2 | 7.2 | 20.1 | 48.8 | 18.7 | 2.7 | 10.9 | 40.8 | 7.9 |
| CDS-CZSL* [28] (CVPR'24) | 32.1 | 48.0 | 64.7 | 60.4 | - | - | - | - | 2.6 | 10.9 | 38.2 | 8.0 |
| PLID [3] (ECCV'24) | 30.8 | 46.6 | 67.6 | 55.5 | 7.3 | 20.4 | 49.1 | 18.7 | 2.5 | 10.6 | 39.1 | 7.5 |
| IMAX [17] (TPAMI'25) | 32.3 | 47.5 | 68.4 | 57.3 | 7.6 | 21.4 | 50.2 | 18.6 | 2.6 | 11.2 | 38.7 | 7.9 |
| ClusPro [47] (ICLR'25) | 39.5 | 54.1 | 71.0 | 66.2 | 9.3 | 23.0 | 51.2 | 22.1 | 3.0 | 11.6 | 41.6 | 8.3 |
| **TOMCAT (Ours)** | 43.7 | 57.9 | 74.1 | 65.8 | 8.2 | 21.7 | 49.2 | 21.0 | 4.2 | 14.2 | 45.1 | 10.6 |

