# OpenReview forum: "TOMCAT: Test-time Comprehensive Knowledge Accumulation for Compositional Zero-Shot Learning"
_NeurIPS.cc/2025/Conference — NeurIPS 2025 poster_

### Official Review · Reviewer_PjJh · 2025-06-10

**Clarity:** 2
**Significance:** 2
**Originality:** 3
**Rating:** 3
**Confidence:** 5

**Summary:**

The core contribution of this paper lies in introducing prompt tuning—commonly used in test-time adaptation tasks—into the context of compositional zero-shot learning. By incorporating several off-the-shelf strategies, the authors achieve promising performance improvements.

**Questions:**

1. The task formulation appears to be a straightforward combination of TTA and CZSL, while the introduced modules—VPC and KAM—are directly adopted from the TTA method TDA[1] and DPE[2] with minor modification. The paper offers only minor modifications that resemble superficial tricks, rather than introducing substantive methodological innovations.While I do not object to integrative works of the "A + B" type, I believe that the components drawn from B should at least be carefully adapted to address the specific challenges of A.


2. The article lacks proper citations, e.g., the KAM module has a very high similarity to the DPE[2] in terms of prototype update strategies in the queue. But unfortunately, I did not find any references to them in this part. Alternatively, the authors should clearly articulate the distinctions between the two approaches.


3. Unfairness in the experimental setup: How can the TTA-based strategies be fairly compared with compositional zero-shot learning models, given the fundamental discrepancy in the amount of accessible information? If the goal is to demonstrate the effectiveness of the proposed method, it is essential to include comparisons with other TTA approaches, rather than only comparing with standard CZSL methods that have no access to test data.

[1]Karmanov, Adilbek, et al. "Efficient test-time adaptation of vision-language models." Proceedings of the IEEE/CVF Conference on Computer Vision and Pattern Recognition. 2024.
[2]Zhang, Ce, et al. "Dual prototype evolving for test-time generalization of vision-language models." Advances in Neural Information Processing Systems 37 (2024): 32111-32136.

**Ethical Concerns:**

["NO or VERY MINOR ethics concerns only"]

**Final Justification:**

After reviewing the paper and the authors’ rebuttal, I find the attempt to bridge TTA and CZSL commendable. The motivation for combining these two areas is well-articulated and represents a meaningful direction. However, I believe the methodological changes introduced in adapting TTA to the CZSL setting are relatively modest and lack surprising innovations. As a result, I can only assign a score that reflects a middle-ground evaluation.

**Limitations:**

Yes

**Paper Formatting Concerns:**

No major formatting issues.

**Quality:**

3

**Strengths And Weaknesses:**

The paper conducts extensive experiments and demonstrates consistently strong results across various settings.

---

> ### Author Rebuttal · Authors · 2025-07-30
>
> Dear Reviewer PjJh,
>
> We would like to express our deepest and most sincere gratitude to you for your thoughtful and constructive comments, which have significantly improved the quality of our work. First, we appreciate your recognition of the extensive experiments and strong results of TOMCAT. Below, we provide our detailed response to each of your valuable comments.
>
> ---
>
> Q1: TOMCAT seems to lack substantive methodological innovations.
>
> R1: First, e are sincerely grateful for your thoughtful concern about methodological innovations; however, we cannot fully agree with this comment with our highest respect. Here we seriously address your questions with great care, based on the following reasons:
>
> 1. Let us elaborate on our motivation in more detail. **Our work is the first to pay attention to the issue of label space shift in CZSL, and for the first time use unsupervised test data to adapt to the new label distribution at test time**, which improves performance in both seen and unseen compositions. The entropy loss used during test overcomes the drawback that unseen compositions cannot be included for optimization during training stage.
> We believe that our method, from the angle of label space shift to solve the CZSL problem, will provide significant inspiration for future research in the CZSL community.
>
> 2. Let us clarify the difference between our method and existing TTA models. TDA[2] records historical images without focusing text information; TPS[3] concentrates on shift vectors but fails to use visual information; DPE[4] updates all prototypes by directly adding shift vectors to them **offline** (failing to accumulate knowledge), which treats all categories equally regardless of familiarity.
> Our work overcomes their limitations with visual priority queue, **online** knowledge accumulation modules, and adaptive update weights based on the similarity of test image and multi-modal prototypes. The adaptive update weight is very effective in addressing the challenge in CZSL where some compositions are seen while others are unseen. Therefore, **at methodology level, we refine the TTA methods, and carefully adjust the refined method with novel components to address the challenge of label space shift in CZSL**, instead of “A+B”.
>
> 3. **There are many outstanding and valuable works building upon prior works in TTA and CZSL, instead of starting from scratch.** In TTA task, the core idea of **TPT**[1] is prompt tuning and data augmentation, which come from **CoOp**[5] and **MEMO**[7], respectively; the core component of **TDA**[2]--visual priority queue--comes from **Tip-Adapter**[6]; the core shift vector in **TPS**[3] is proposed by **SSF**[8]; in **DPE**[4], the visual cache and shift vector are inspired by **TDA**[2] and **TPS**[3]. In CZSL, the proposal of **CSP**[9] is to introduce prompt tuning to CZSL field from **CoOp**[5], inspiring many prompt-tuning based works; the core idea of **Troika**[10]--disentanglement--has shown its effectiveness in non-CLIP-based models[11,12]; the main component GNN in **GIPCOL**[13] is inspired by **Co-CGE**[14].
> **The usage of prompt tuning technology in CZSL has in turn contributed to the development of prompt-tuning itself[10,15].** **Similarly, our work provides a new test-time approach to address CZSL, and it will in turn promote the development of TTA.**
> Similar to our work, these high-quality methods have **applied a particular technique to another practical problems after refinement in a adaptive and flexible manner and achieved strong results**. We cannot deny the valuable contributions they have made to the community.
>
> 4. After rebuttal, we will revise our paper following your valuable suggestions--**further clarify the novelty and the differences with related methods in more detail**. In addition, considering that Reviewers hxuG, uhct and 45LQ have agreed that our method is novel (new) with good merits, we think our work presents an important insight for advancing CZSL, and believe that our research is worth sharing with the community.
>
> ---
>
> Q2: Unclear citation.
>
> R2: We are very sorry that the citations in our paper have caused your misunderstandings. First, we have cited all methods and code inspiring our work in Related Work and Appendix J. In addition, we have made appropriate citations to Tip-Adapter[6], AdapterFormer[16], TDA[2], and CoOp[5] in Method Section; and we will appropriately add more citations of models such as TPS[3] and DPE[4] in this section.
>
> Thank you for your kind reminder regarding the citations, which are now included in the current manuscript.
>
> ---
>
> Q3: Unfairness in the experimental setup.
>
> R3: To alleviate the reviewer’s concerns about fairness, we provide experimental results under three different settings:
> **(1) comparing different CZSL methods using test-time TOMCAT on CZSL datasets** (disentanglement-based models like Troika are not compatible with TOMCAT due to disentangling multi-prototypes) in Table 1;
> **(2) comparing TOMCAT with TTA methods on standard TTA classification datasets** (e.g., TPT[1], TDA[2], TPS[3], and DPE[4]) in Table 2.
> **(3) comparing TOMCAT and different TTA methods based on the Base Model on CZSL datasets** in Table 3.
> The results show that TOMCAT performs better than baselines with fairness in different fair settings.
>
> | Method            | UT-Zappos |       |       |        | MIT-States |       |       |        | C-GQA |       |       |        |
> |-------------------|------------|-------|-------|--------|-----------|-------|-------|--------|-------|-------|-------|--------|
> |                   | AUC        | HM    | Seen  | Unseen | AUC       | HM    | Seen  | Unseen | AUC   | HM    | Seen  | Unseen |
> | CSP[9]               | 31.24      | 46.55 | 60.9  | 62.62  | 19.13     | 25.92 | 46.68 | 49.67  | 5.85  | 19.45 | 27.54 | 26.92  |
> | CSP[9] + TOMCAT        | 32.26      | 49.27 | 64.32 | 57.69  | 19.48     | 26.77 | 49.96 | 49.96  | 6.09  | 19.8  | 28.88 | 26.75  |
> | Base Model        | 43.57      | 55.54 | 68.72 | **74.30**   | 22.12     | 38.97 | 49.50  | 52.61  | 10.73 | 27.92 | 36.63 | 24.39  |
> | Base Model + TOMCAT | **48.31**      | **60.18** | **74.49** | 72.77  | **22.55**     | **39.45** | **50.22** | **52.95**  | **11.05** | **28.48**  | **37.66**  | **34.18**  |
>
> Table 1. Comparison of the results of models with and without TOMCAT on three CZSL datasets.
>
> | Method | ImageNet | ImageNet V2 | Flower102 | DTD   |
> |--------|----------|-------------|------------|-------|
> | TPT[1]    | 69.54    | 63.95       | 69.71      | 46.93 |
> | TDA[2]    | 69.51    | 64.67       | 71.42      | 47.40 |
> | TPS[3]    | 71.45    | 64.91       | 71.54      | 50.47 |
> | DPE[4]    | 71.91    | **65.44**       | 75.07      | 54.20  |
> | TOMCAT | **72.09**   | 64.83       | **75.46**      | **56.03** |
>
> Table 2. Comparison of the top-1 classification accuracy among TOMCAT with TTA methods on four standard TTA classification datasets.
>
> | Method            | UT-Zappos |       |       |        | MIT-States |       |       |        | C-GQA |       |       |        |
> |-------------------|------------|-------|-------|--------|-----------|-------|-------|--------|-------|-------|-------|--------|
> |                   | AUC        | HM    | Seen  | Unseen | AUC       | HM    | Seen  | Unseen | AUC   | HM    | Seen  | Unseen |
> | online-TPS[3] | 44.59 | 57.29 | 71.26 | 71.44 | 22.21 | 39.09 | 49.62 | 52.8  | 10.84 | 27.87 | 37.51 | 33.65 |
> | TDA[2]        | 41.57 | 54.61 | 69.01 | 70.6  | 22.01 | 88.71 | 49.5  | 52.86 | 9.41  | 25.3  | 37.25 | 30.33 |
> | DPE[4]       | 43.32 | 55.97 | 68.49 | **74.17** | 22.02 | 38.9  | 49.24 | 52.78 | 10.69 | 27.88 | 36.57 | 34    |
> | TOMCAT | **48.31**      | **60.18** | **74.49** | 72.77  | **22.55**     | **39.45** | **50.22** | **52.95**  | **11.05** | **28.48**  | **37.66**  | **34.18**  |
>
> Table 3. Comparison of TOMCAT and different TTA methods based on the Base Model on CZSL datasets.
>
> ---
>
> This concludes our response to each of your comments. Once again, we wholeheartedly and sincerely express our gratitude to your time and thoughtful consideration. We would be happy to further discuss if you have any remaining concerns.
>
> Best regards,
>
> Authors.
>
> &nbsp;
>
> [1] Manli Shu, et al. Test-time prompt tuning for zero-shot generalization in vision-language models. NeurIPS 2022.
>
> [2] Adilbek Karmanov, et al. Efficient test-time adaptation of vision-language models. CVPR 2024.
>
> [3] Just shift it: Test-time prototype shifting for zero-shot generalization with vision-language models. WACV 2025.
>
> [4] Ce Zhang, et al. Dual prototype evolving for test-time generalization of vision-language models. NeurIPS 2024.
>
> [5] Kaiyang Zhou, et al. Conditional prompt learning for vision-language models. CVPR 2022.
>
> [6] Renrui Zhang, et al. Training-free adaption of clip for few-shot classification. ECCV 2022.
>
> [7] Marvin Zhang, et al. MEMO: test time robustness via adaptation and augmentation. arXiv preprint 2021.
>
> [8] Dongze Lian, et al. Scaling & Shifting Your Features: A New Baseline for Efficient Model Tuning. NeurIPS 2022.
>
> [9] Nihal V. Nayak, et al. Learning to compose soft prompts for compositional zero-shot learning. ICLR 2023.
>
> [10] Siteng Huang, et al. Troika: Multi-path cross-modal traction for compositional zero-shot learning. CVPR 2024.
>
> [11] Nirat Saini, et al.  Disentangling visual embeddings for attributes and objects. CVPR 2022.
>
> [12] Shaozhe Hao, et al. Learning attention as disentangler for compositional zero shot learning. CVPR 2023.
>
> [13] Guangyue Xu, et al. GIPCOL: Graph-Injected Soft Prompting for Compositional Zero-Shot Learning. WACV 2024.
>
> [14] Massimiliano Mancini, et al. Learning graph embeddings for open world compositional zero-shot learning. TPAMI 2022.
>
> [15] Wentao Bao, et al. Prompting language-informed distribution for compositional zero-shot learning. ECCV 2024.
>
> [16] Shoufa Chen, et al. Adaptformer: Adapting vision transformers for scalable visual recognition. NeurIPS 2022.

---

> > ### Comment · Reviewer_PjJh · 2025-08-05
> >
> > Thank you for the detailed rebuttal. I appreciate the authors' thoughtful discussion on the novelty aspect, particularly the motivation behind combining TTA with CZSL. The perspective is compelling, and it has helped me better recognize the potential of this combination.
> >
> > However, I still believe that the methodological contribution on the TTA side is relatively minor. While the idea is creative, the technical novelty appears limited. As such, while I may slightly adjust my score to reflect the clarified motivation, I do not believe a significant increase is warranted.

---

> ### Author Response · Authors · 2025-08-05
> **Author Response to Reviewer PjJh by Authors**
>
> Dear Reviewer PjJh,
>
> We are wholeheartedly grateful for your thoughtful and constructive feedback. I would like to offer a clearer explanation here.
>
> - You have mentioned the methodological contribution on the TTA side. However, as part of the paper title "for CZSL", we aim to **provide a novel test-time approach for CZSL**, **with less attention on TTA**. TOMCAT provides a novel direction using unsupervised test data to address label space shift and improve performance, **which is not previously considered in CZSL.**
> - **Our focus has always been CZSL problem rather than TTA. A reviewer should not assess the value and contribution of our work based on a field that we did not aim to address.**
> - **We believe our work, like CSP[1]-- which has made prompt tuning of CLIP become a mainstream technique for CZSL-- will inspire a new wave of test-time adaptation based approaches in CZSL.** And these works, in turn, will promote the development of TTA.
> - Regarding technical novelty in TTA, we have to say TTA is a relatively mature task, where major architectural breakthroughs have become increasingly rare.
> In such a context, recent advances, including replacing augmentation with diffusion generation[2], changing probability averaging of augmented views to voting[3], exploiting visual cache [4], using LoRA visual adapter[5], leveraging Gaussian Mixture Model (GMM)[6], and introducing a regularization term[7], rely on carefully designed incremental innovations to advance the state of the art. In addition, we made relatively substantial modifications to better address the challenges of CZSL at the methodology level.
>
> Based on the points above, **we believe that our work is worth sharing with the community**. Please give us an opportunity to explain if you have remaining concerns about our novelty. No matter what score is ultimately assigned to our paper, we are truly and deeply grateful for your valuable and constructive feedback.
>
> With great respect,
>
> Authors.
>
> &nbsp;
>
> [1] Nihal V. Nayak, et al. Learning to compose soft prompts for compositional zero-shot learning. ICLR 2023.
>
> [2] Chun-Mei Feng, et al. Diverse Data Augmentation with Diffusions for Effective Test-time Prompt Tuning. ICCV 2023.
>
> [3] Matteo Farina, et al. Frustratingly Easy Test-Time Adaptation of Vision-Language Models. NeurIPS 2024.
>
> [4] Adilbek Karmanov, et al. Efficient test-time adaptation of vision-language models. CVPR 2024.
>
> [5] Raza Imam, et al. Test-Time Low Rank Adaptation via Confidence Maximization for Zero-Shot Generalization of Vision-Language Models. WACV 2025.
>
> [6] Qiyuan Dai, et al. Free on the Fly: Enhancing Flexibility in Test-Time Adaptation with Online EM. CVPR 2025.
>
> [7] Ashshak Sharifdeen et al. O-TPT: Orthogonality Constraints for Calibrating Test-time Prompt Tuning in Vision-Language Models. CVPR 2025.

---

### Official Review · Reviewer_45LQ · 2025-06-21

**Clarity:** 3
**Significance:** 3
**Originality:** 3
**Rating:** 4
**Confidence:** 4

**Summary:**

This paper introduces TOMCAT, a novel approach for addressing label distribution shift in Compositional Zero-Shot Learning (CZSL) by dynamically updating multimodal prototypes at test time. The method demonstrates significant improvements in performance on multiple benchmark datasets, particularly in the open-world setting.

**Questions:**

NO

**Ethical Concerns:**

["NO or VERY MINOR ethics concerns only"]

**Final Justification:**

Thanks for the author's reply. I tend to give it a positive score but not a completely acceptable rate.

**Quality:**

3

**Strengths And Weaknesses:**

Strengths:
1. Innovation: TOMCAT introduces a novel approach to address the label distribution shift in Compositional Zero-Shot Learning (CZSL) by dynamically updating multimodal prototypes during testing, making it a significant advancement in the field.
2. Writing: The paper is well-written and clearly structured, making the methodology easy to understand and follow. The experiments are presented effectively, and the results are clearly communicated, enhancing the overall readability.

Weaknesses:
1. Comparison with Recent Methods: The paper mostly compares TOMCAT to methods from a year ago or older. Given the fast-moving nature of the field, it would be beneficial to compare TOMCAT against more recent methods. For example, the work from 2024 or 2025 in test-time adaptation and zero-shot learning would provide a more compelling argument for TOMCAT's advantages.
2. Resource and Latency Analysis: While the authors mention deferred backpropagation to reduce latency during testing, there is little discussion on the scalability and resource consumption of TOMCAT in more complex, real-world environments. Further analysis on memory usage and computational demands, especially for large-scale deployments, would be valuable.
3. Limited Discussion of Limitations: The paper does not sufficiently discuss the limitations of the approach. For example, how well does TOMCAT handle extreme label distribution shifts or ambiguous compositions in real-world settings? The model’s performance on datasets with noise or extremely variable compositions should be further explored.

---

> ### Author Rebuttal · Authors · 2025-07-30
>
> Dear Reviewer 45LQ,
>
> We would like to express our deepest and most sincere gratitude to you for your constructive comments, which significantly improve the quality of our work. First, we appreciate your recognition of our work: (1) TOMCAT introduces a novel approach to address the label distribution shift in CZSL. (2) The paper is well-written and clearly structured.
>
> For the three weaknesses raised, we are sincerely sorry for putting them in the Appendix due to the page limitation. Specifically, comparison with more baselines is presented in Appendix E (Table 14 and Table 15); Resource and Latency Analysis is provided in Appendix G (Table 13); Discussion of Limitations is presented in Appendix H. Now we would like to provide a more detailed response to each comment below:
>
> ---
>
> **W1:** Comparison with Recent Methods.
>
> **R1:** In Table 14 and Table 15 in the Appendix, we compare many recent and prominent CZSL approaches, including six methods proposed in 2024 and 2025. After rebuttal, we will add more baselines in the manuscript, including CoP[1](ICME 2024), CAGC[2](TNNLS 2025), CLPS[3](TMM 2025), and more.
>
> In addition, you have mentioned that it is better to compare with test-time adaptation (TTA) baselines, so **we compare TOMCAT with them on CZSL datasets** in Table 1. Also, **we compare TOMCAT with them under standard TTA setting** in Table 2. TOMCAT outperforms them under CZSL and TTA setting, indicating that TOMCAT has great potential in both CZSL and TTA tasks.
>
> | Method            | UT-Zappos |       |       |        | MIT-States |       |       |        | C-GQA |       |       |        |
> |-------------------|------------|-------|-------|--------|-----------|-------|-------|--------|-------|-------|-------|--------|
> |                   | AUC        | HM    | Seen  | Unseen | AUC       | HM    | Seen  | Unseen | AUC   | HM    | Seen  | Unseen |
> | online-TPS[5] | 44.59 | 57.29 | 71.26 | 71.44 | 22.21 | 39.09 | 49.62 | 52.8  | 10.84 | 27.87 | 37.51 | 33.65 |
> | TDA[6]        | 41.57 | 54.61 | 69.01 | 70.6  | 22.01 | 88.71 | 49.5  | 52.86 | 9.41  | 25.3  | 37.25 | 30.33 |
> | DPE[7]       | 43.32 | 55.97 | 68.49 | **74.17** | 22.02 | 38.9  | 49.24 | 52.78 | 10.69 | 27.88 | 36.57 | 34    |
> | TOMCAT | **48.31**      | **60.18** | **74.49** | 72.77  | **22.55**     | **39.45** | **50.22** | **52.95**  | **11.05** | **28.48**  | **37.66**  | **34.18**  |
>
> Table 1. Comparison of TOMCAT and different TTA methods based on the Base Model on CZSL datasets.
>
> | Method | ImageNet | ImageNet V2 | Flower 102 | DTD   |
> |--------|----------|-------------|------------|-------|
> | TPT[4]    | 69.54    | 63.95       | 69.71      | 46.93 |
> | TPS[5]    | 71.45    | 64.91       | 71.54      | 50.47 |
> | TDA[6]   | 69.51    | 64.67       | 71.42      | 47.40 |
> | DPE[7]    | 71.91    | 65.44       | 75.07      | 54.20  |
> | TOMCAT | **72.09**   | **64.83**       | **75.46**      | **56.03** |
>
> Table 2. Comparison of the top-1 classification accuracy on four standard TTA classification datasets.
>
> ---
>
> **W2:** Resource and Latency Analysis.
>
> **R2:** As shown in Table 3 (Table 13 in the Appendix G), we present the time, decay, and GPU memory occupation of the base model under standard test setting and TOMCAT. **Our TOMCAT only performs minimal additional decay and computational resources at test time.**
>
> |Dataset | Time |  | Latency |  | GPU Memory |  |
> |------------|--------|------------|--------|------------|--------|------------|
> |            | TOMCAT | w/o TOMCAT | TOMCAT | w/o TOMCAT | TOMCAT | w/o TOMCAT |
> | UT-Zappos  | 47ms   | 36ms       | 35ms   | 32ms       | 4044MB | 3801MB     |
> | MIT-States | 145ms  | 70ms       | 112ms  | 37ms       | 4180MB | 4175MB     |
>
> Table 3. Comparison of time, decay, and memory between TOMCAT and the base model (w/o TOMCAT) on UT-Zappos and MIT-States. Time means that average testing time across all samples. Latency indicates time from input to prediction output for per sample (excluding backpropagation). Memory represents the GPU memory occupied by the model during testing.
>
> ---
>
> **W3:** Discussion of limitation.
>
> **R3:** As presented in Appendix H, two potential limitations of TOMCAT are identified: (1) The prediction entropy incorporates unseen compositions into optimization to adapt to the new distribution, but it may cause the model to make incorrect predictions with greater confidence. (2) Although many measures have been taken to reduce latency, the backpropagation still increases the time overhead at test time. We hope that future research can focus more on the label distribution shift in CZSL and better address these two limitations.
>
> ---
>
> The above summarizes our response to your concerns. Once again, we sincerely appreciate your constructive and thoughtful comments. And we would be happy to further discuss if there are any remaining concerns.
>
> Best regards,
>
> Authors.
>
> &nbsp;
>
> [1] Tian Zhang, et al. Learning Conditional Prompt for Compositional Zero-Shot Learning. ICME 2024.
>
> [2] Yang Liu, et al. Concept-Aware Graph Convolutional Network for Compositional Zero-Shot Learning. TNNLS 2025.
>
> [3] Han Jiang. Compact Latent Primitive Space Learning for Compositional Zero-Shot Learning. TMM 2025.
>
> [4] Manli Shu, et al.  Test-time prompt tuning for zero-shot generalization in vision-language models. NeurIPS 2022.
>
> [5] Elaine Sui, et al. Just shift it: Test-time prototype shifting for zero-shot generalization with vision-language models. WACV 2025.
>
> [6] Adilbek Karmanov, et al. Efficient test-time adaptation of vision-language models. CVPR 2024.
>
> [7] Ce Zhang, et al. Dual prototype evolving for test-time generalization of vision-language models. NeurIPS 2024.

---

> ### Author Response · Authors · 2025-08-06
> **Author Response to Reviewer 45LQ by Authors**
>
> Dear Reviewer 45LQ,
>
> Thanks for your time and thoughtful consideration. We notice that you have posted the Acknowledgement without discussion, so we are not sure whether we have addressed your concerns. We know that the reviewers are busy and their time is valuable, so we briefly outline your concerns and our corresponding response to save your time:
>
> - **Comparison with Recent Methods**：We have compared many recent baselines including six methods proposed in 2024 and 2025. We will add more recent baselines in the paper.
>
> - **Resource and Latency Analysis**: Our TOMCAT only performs minimal additional decay and computational resources at test time in Table 3 in the Rebuttal.
>
> - **Discussion of Limitations**: TOMCAT may cause incorrect predictions with greater confidence. And it still increase minimal time overhead although many measures have been taken to reduce latency.
>
> **In addition, the More Comparison, Resource and Latency Analysis, and Discussion of Limitation are also provided in Appendix E, G, and H.** We would be very grateful for the opportunity to discuss any remaining or additional concerns that you may have. Thank you again for your time and effort in proposing useful suggestions.
>
> With great respect,
>
> Authors.

---

### Official Review · Reviewer_uhct · 2025-06-26

**Clarity:** 3
**Significance:** 2
**Originality:** 3
**Rating:** 4
**Confidence:** 4

**Summary:**

Current Compositional Zero-Shot Learning (CZSL) methods often struggle with performance declines during testing due to shifts in label space distribution resulting from unseen attribute-object combinations. This paper proposes a novel approach that collects extensive knowledge from unsupervised data in both textual and visual domains to update multi-modal prototypes at test time. An adaptive update weight is introduced to control prototype adjustments, enhancing the model's adaptability to distribution shifts. A dynamic priority queue is also employed to store high-confidence images, allowing the model to leverage visual knowledge from historical data for better inference. To maintain semantic consistency, textual and visual prototypes are aligned through multimodal collaborative representation learning. Extensive experiments show the method surpasses existing ones on four challenging benchmark datasets, under both closed-world and open-world conditions.

**Questions:**

1. Does the performance on test data improve progressively with the inference order? In other words, would the prediction results be relatively better for data that comes later in the sequence?
2 . Will the performance increase if the method uses BLIP2 or a more powerful foundational model?

**Ethical Concerns:**

["NO or VERY MINOR ethics concerns only"]

**Final Justification:**

Thank the authors for the detailed response. The proposed method of the paper can improve upon existing methods on the CZSL dataset and also demonstrates outstanding performance on standard TTA classification datasets. Therefore, I consider raising my score and tend to accept this work.

**Limitations:**

yes

**Paper Formatting Concerns:**

The order of presentation can be adjusted appropriately. For instance, the variable "t" mentioned in Eqn(2) is defined in Eqn(5), which is not friendly to readers.

**Quality:**

2

**Strengths And Weaknesses:**

[s1] The proposed TOMCAT can accumulate multimodal knowledge from unlabeled data and updates prototypes at test time to bridge the label distribution shift.

[s2] A priority queue stores historical information and adaptively updates multimodal prototypes.

[s3]  The performance of TOMCAT on four benchmark datasets is the best in both closed-world and open-world settings.

[w1] The main issue is that the evaluation is not entirely fair. The evaluation of existing CZSL methods does not rely on test data. However, the proposed method utilizes historical data to obtain visual and textual information through CLIP. Alternatively, from one perspective, it leverages CLIP to annotate a portion of the data, which is not as fair in comparison to previous approaches. It might be the primary reason for the significant performance improvement observed in this paper.

---

> ### Author Rebuttal · Authors · 2025-07-30
>
> Dear Reviewer uhct,
>
> We would like to wholeheartedly and sincerely thank you for your constructive comments, which we found immensely helpful. First, we appreciate your recognition of our TOMCAT: (1) This paper proposes a novel approach with some useful components. (2) Our method achieves SOTA performance across four datasets. We now provide detailed explanations on each of your concerns.
>
> ---
>
> **W1:** The evaluation is not entirely fair.
>
> **Q1:** First, we sincerely thank you for providing the insightful concern about fairness of comparison, which is understandable, but we want to explain our comparison is reasonably fair, based on the following reasons:
>
> 1. Compared with the existing methods, **there is no complex disentanglement or other modules for training in TOMCAT.** However, TOMCAT still outperforms them.
> 2. Since the model simulates the scenario where a user submits a single test image **without accessing its ground-truth label**, we consider this a fair evaluation.
> 3. Existing methods simply discard the test data after evaluation, which we believe is a big waste. Instead, we propose to make use of this unlabeled data.
> 4. You have provided a very insightful and profound comment--it seems that we leverage CLIP to annotate a portion of the data at test time. However, we have to emphasize that **CLIP is not an external auxiliary helper--it is the core predictive model in current mainstream methods** (such as all baselines in Table 1 in the paper). In standard evaluation setting, it has to predict an image; TOMCAT just uses the prediction.
> On the other hand, **adapting the model using its own predictions follows a self-supervised paradigm** (e.g., BLIP[7]), which is a widely acceptable and reasonable practice.
> 5. To further alleviate the reviewer’s concerns about fairness, we provide experimental results under three different settings:
> **(1) comparing different CZSL methods using test-time TOMCAT on CZSL datasets** (disentanglement-based models like Troika are not compatible with TOMCAT due to disentangling multi-prototypes) in Table 1;
> **(2) comparing TOMCAT with TTA methods on standard TTA classification datasets** (e.g., TPT[4], TPS[5], TDA[2], and DPE[6]) in Table 2.
> **(3) comparing TOMCAT and different TTA methods based on the Base Model on CZSL datasets** in Table 3.
> The results show that TOMCAT performs better than baselines with fairness in different settings.
>
> | Method            | UT-Zappos |       |       |        | MIT-States |       |       |        | C-GQA |       |       |        |
> |-------------------|------------|-------|-------|--------|-----------|-------|-------|--------|-------|-------|-------|--------|
> |                   | AUC        | HM    | Seen  | Unseen | AUC       | HM    | Seen  | Unseen | AUC   | HM    | Seen  | Unseen |
> | CSP[3]               | 31.24      | 46.55 | 60.9  | 62.62  | 19.13     | 25.92 | 46.68 | 49.67  | 5.85  | 19.45 | 27.54 | 26.92  |
> | CSP[3] + TOMCAT        | 32.26      | 49.27 | 64.32 | 57.69  | 19.48     | 26.77 | 49.96 | 49.96  | 6.09  | 19.8  | 28.88 | 26.75  |
> | Base Model        | 43.57      | 55.54 | 68.72 | **74.30**   | 22.12     | 38.97 | 49.50  | 52.61  | 10.73 | 27.92 | 36.63 | 24.39  |
> | Base Model + TOMCAT | **48.31**      | **60.18** | **74.49** | 72.77  | **22.55**     | **39.45** | **50.22** | **52.95**  | **11.05** | **28.48**  | **37.66**  | **34.18**  |
>
> Table 1. Comparison the results of CSP and the Base Model with and without TOMCAT on three CZSL datasets.
>
> | Method | ImageNet | ImageNet V2 | Flower102 | DTD   |
> |--------|----------|-------------|------------|-------|
> | TPT[4]    | 69.54    | 63.95       | 69.71      | 46.93 |
> | TPS[5]    | 71.45    | 64.91       | 71.54      | 50.47 |
> | TDA[2]    | 69.51    | 64.67       | 71.42      | 47.40 |
> | DPE[6]    | 71.91    | **65.44**       | 75.07      | 54.20  |
> | TOMCAT | **72.09**   | 64.83       | **75.46**      | **56.03** |
>
> Table 2. Comparison of the top-1 classification accuracy among TOMCAT with TTA methods on four standard TTA classification datasets.
>
> | Method            | UT-Zappos |       |       |        | MIT-States |       |       |        | C-GQA |       |       |        |
> |-------------------|------------|-------|-------|--------|-----------|-------|-------|--------|-------|-------|-------|--------|
> |                   | AUC        | HM    | Seen  | Unseen | AUC       | HM    | Seen  | Unseen | AUC   | HM    | Seen  | Unseen |
> | online-TPS[5] | 44.59 | 57.29 | 71.26 | 71.44 | 22.21 | 39.09 | 49.62 | 52.8  | 10.84 | 27.87 | 37.51 | 33.65 |
> | TDA[2]        | 41.57 | 54.61 | 69.01 | 70.6  | 22.01 | 88.71 | 49.5  | 52.86 | 9.41  | 25.3  | 37.25 | 30.33 |
> | DPE[6]       | 43.32 | 55.97 | 68.49 | **74.17** | 22.02 | 38.9  | 49.24 | 52.78 | 10.69 | 27.88 | 36.57 | 34    |
> | TOMCAT | **48.31**      | **60.18** | **74.49** | 72.77  | **22.55**     | **39.45** | **50.22** | **52.95**  | **11.05** | **28.48**  | **37.66**  | **34.18**  |
>
> Table 3. Comparison of TOMCAT and different TTA methods based on the Base Model on CZSL datasets.
>
> ---
>
> **Q1:** The impact of different inference order.
>
> **R2:** Generally, the images coming later are predicted relatively better due to the knowledge accumulation, where the model is anticipated to evolve better during test. As shown in Figure 5 in the paper, TOMCAT start obviously higher than base model from about 1/3 point of the sequence. However, we focus more on the overall performance of the model on the dataset.
>
> In addition, Table 4 provides three different ordering, from which we observe that **there is variance among different ordering, but they are not significant.** We attribute this to the fact that adaptive weights and the regularization item of multimodal collaborative representation learning enhance the stability of the model, allowing the model to adaptively determine the extent to which the prototypes are updated.
>
> | Order | UT-Zappos |       |       |        | MIT-States |       |       |        | C-GQA |       |       |        |
> |-------|------------|-------|-------|--------|-----------|-------|-------|--------|-------|-------|-------|--------|
> |       | AUC        | HM    | Seen  | Unseen | AUC       | HM    | Seen  | Unseen | AUC   | HM    | Seen  | Unseen |
> | 1     | 48.31      | 60.18 | 74.49 | 72.77  | 22.55     | 39.45 | 50.22 | 52.95  | 11.05 | 28.48  | 37.66  | 34.18   |
> | 2     | 48.52      | 60.82 | 74.64 | 72.95  | 22.46     | 39.33 | 49.84 | 53.05  | 11.38 | 28.93 | 38.44 | 34.18  |
> | 3     | 46.44      | 58.72 | 72.63 | 72.03  | 22.31     | 39.02 | 49.62 | 53.09  | 11.04 | 28.51 | 38.09 | 33.48  |
>
> Table 4. The reusults of different test ordering on three CZSL datasets.
>
> ---
>
> **Q2:** Comparison with a more powerful foundational model like BLIP-2.
>
> **R3:** At its core, CZSL is a representation learning task rather than a generative task; hence, MLLMs like BLIP-2 and LLaVA, whose behaviour is hard to control, are not suitable for this purpose.
>
> However, we still conduct this experiment following your suggestions using MLLMs. BLIP-2 exhibits limited instruction-following capability and can only generate a caption of the whole image such as "a pair of knives with a box next to them” instead of the composition "folded knife". Therefore, we use two stronger MLLMs, LLaVA v1.5[8] and InternVL-3[1]. **Table 5 shows that TOMCAT outperforms them on three datasets in CZSL, indicating CLIP-based models are more suitable for representation tasks** (e.g., CZSL, recognition, and retrieval).
>
> | Method     | UT-Zappos     |             |                  |            | MIT-States |             |                  |       |
> |------------|---------------|-------------|------------------|------------|---------------------|-------------|------------------|-------|
> |            | Attribute Acc↑ | Object Acc↑ | Composition Acc↑ | Time (ms)↓ | Attribute Acc↑       | Object Acc↑ | Composition Acc↑ | Time (ms)↓ |
> | LLaVA v.15[8] | 2.13          | 10.17       | 0.32             | 463        | 1.06                | 21.77       | 5.72             | 857   |
> | InternVL-3[1] | 8.77          | 23.71       | 2.06             | 5770       | 8.53                | 48.28       | 4.43             | 728   |
> | TOMCAT     | **58.54**         | **77.45**       | **44.88**            | **47**        | **41.00**                  | **56.58**       | **30.92**            | **145**   |
>
> Table 5. The classification accuracy of TOMCAT and two MLLMs. Since MLLMs cannot generate classification probability, we report the top-1 accuracy instead of AUC and HM.
>
> ---
>
> **PFC:** The order of presentation can be adjusted appropriately.
>
> **R4:** We are sincerely sorry to mislead the readers due to our writing order. We will carefully adjust the order of our presentation following your valuable suggestion after rebuttal.
>
> ---
>
> The above provides our responses to your comments. Once again, we are grateful for your time and thoughtful feedback. We would be happy to further discuss if any concerns remain.
>
> Best regards,
>
> Authors.
>
> &nbsp;
>
> [1] Jinguo Zhu, et al. Internvl3: Exploring advanced training and test-time recipes for open-source multimodal models. arXiv preprint 2025.
>
> [2] Adilbek Karmanov, et al. Efficient test-time adaptation of vision-language models. CVPR 2024.
>
> [3] Nihal V. Nayak, et al. Learning to compose soft prompts for compositional zero-shot learning. ICLR 2023.
>
> [4] Manli Shu, et al. Test-time prompt tuning for zero-shot generalization in vision-language models. NeurIPS 2022.
>
> [5] Elaine Sui, et al. Just shift it: Test-time prototype shifting for zero-shot generalization with vision-language models. WACV 2025.
>
> [6] Ce Zhang, et al. Dual prototype evolving for test-time generalization of vision-language models. NeurIPS 2024.
>
> [7] Junnan Li, et al. BLIP: Bootstrapping Language-Image Pre-training for Unified Vision-Language Understanding and Generation. ICML 2022.
>
> [8] Haotian Liu, et al. Improved baselines with visual instruction tuning. CVPR 2024.

---

> > ### Comment · Reviewer_uhct · 2025-08-05
> >
> > Thank the authors for the detailed response. The supplemented experimental results show that the proposed method improves upon existing methods on the CZSL dataset and also demonstrates outstanding performance on standard TTA classification datasets.

---

> ### Author Response · Authors · 2025-08-05
> **Response to Reviewer uhct by Authors**
>
> Dear Reviewer uhct,
>
> We are truly grateful for your feedback and your recognition of the performance of our method in supplementary experiments. Nonetheless, our experiments are primarily designed to **address your concern about fairness**, and secondarily to examine the potential impact of test order and the performance of MLLMs like BLIP-2.
>
> We would be happy to discuss them if there are any remaining or additional concerns. If all your concerns have been addressed, we would also appreciate it if you could kindly let us know.
>
> Once again, thanks for your valuable time and feedback.
>
> With great respect,
>
> Authors.

---

### Official Review · Reviewer_4ZG6 · 2025-07-02

**Clarity:** 2
**Significance:** 2
**Originality:** 3
**Rating:** 5
**Confidence:** 3

**Summary:**

Authors present TOMCAT, a compositional zero-shot learning approach that leverages test-time knowledge accumulation to identify novel compositions of attributes and objects not seen during training. Authors learn multi-modal prototypes for each combination of attributes and categories that are updated during inference using a confidence-based dynamic priority queue. Authors demonstrate that their proposed approach works better than their base model over time (e.g. as class prototypes are refined throughout inference), and significantly outperforms prior work on four datasets in both closed-world and open-world settings.

**Questions:**

- Authors train the base model on in-domain data and see examples of typical compositions during training. How is the proposed method zero-shot? Although it may generalize to novel compositions, it has already seen the domain, making it difficult to evaluate how well the model is generalizing. Instead, one can consider using the CLIP model as the base architecture and demonstrate compositional generalization without any training.
- How does the ordering of test images impact final performance? Do we expect to see significant variance?
- What is the impact of the size of the label space C on the model performance?
- Authors highlight latency during inference time as a primary consideration in the design of their method. What is the latency of this method compared to prior work?

**Ethical Concerns:**

["NO or VERY MINOR ethics concerns only"]

**Final Justification:**

Authors have sufficiently addressed my questions and provide strong experimental evidence to support their claims. I would recommend accepting this paper.

**Limitations:**

Yes, authors address the limitations of their work in the supplement.

**Quality:**

3

**Strengths And Weaknesses:**

Strenghts
- Strong Results. Authors demonstrate that that their proposed approach significantly outperforms prior work.
- Extensive Baseline Comparisons. Authors compare against many baselines (and include even more results in the supplement).
- Informative Visuals. The visuals help explain the core components of the method, making it easy to understand the proposed approach.

Weaknesses
- Unclear Writing. The writing is unclear at times, making it difficult to understand the fine-grained details. Authors naming each individual component of their method makes it more confusing to follow their approach.
- Limited Open-World Benchmark. Although the total number of potential attribute-object pairs is large (especially for the C-GQA) benchmark, many of the attribute-object pairs are not realistic, limiting the total size of the "open-world". This textual bias may make it easier to ignore certain attribute-object pairs. Instead, real attribute-object pairs should be sequestered during training.
- Bias Towards Compositional Datasets. The proposed approach seemingly has a bias towards compositional objects. Although I understand that this is the primary focus of the paper, it is unlikely that real-world scenarios will only exclusively have compositional objects. This paper can be strengthened by evaluating on standard classification datasets to validate that the method does not improve compositional performance at the cost of non-compositional classification accuracy.

---

> ### Author Rebuttal · Authors · 2025-07-30
>
> Dear Reviewer 4ZG6,
>
> We are truly and profoundly grateful to you for your insightful suggestions and careful evaluation, which have been invaluable in enhancing our work. First, we appreciate your recognition of our work: (1) TOMCAT has strong results. (2) We provide extensive comparisons with many baselines and informative visual. We now provide detailed response to your comments.
>
> ---
>
> **W1:** Unclear writing.
>
> **R1:** We sincerely apologize for the confusion caused by our writing and naming conventions. We will take the reviewer's suggestions seriously and refine our writing and naming practices in this paper and our future works after rebuttal. We would be happy to explain if you point out the unclear points. In addition, we promise to immediately release the code after the final decision to facilitate readers’ understanding.
>
> ---
>
> **W2:** Limited Open-World Benchmark caused by infeasible compositions.
>
> **R2:** We use a post-training feasibility calibration (computed by GloVE[1] word embedding) to filter out part of infeasible compositions following CSP[2], Troika[3] and PLID[4]. During training, we sequester the real attribute-object pairs. In the open-world setting, we do not sequester them at test time. Instead, **the model is required to infer them autonomously, assigning lower prediction probabilities to infeasible combinations.** We believe this setting is more challenging than assuming a predefined set of feasible compositions (closed-world setting).
>
> ---
>
> **W3:** Bias Towards compositional datasets.
>
> **R3:** Our method is proposed to overcome the label space shift caused by the unseen compositions recombined by attributes and objects at test time. However, as you say, a robust model does not improve compositional performance at the cost of standard classification accuracy.
>
> Therefore, on the one hand, following your suggestion, **we evaluate TOMCAT on four standard datasets (two ImageNet datasets and two fine-grained datasets) in Table 1**. On the other hand, **we report the pure object classification accuracy on CZSL datasets in Table 2.** Both of the results demonstrate that our TOMCAT performs well on standard object recognition too.
>
> | Method | ImageNet | ImageNet V2 | Flower 102 | DTD   |
> |--------|----------|-------------|------------|-------|
> | TPT[7]    | 69.54    | 63.95       | 69.71      | 46.93 |
> | TPS[8]    | 71.45    | 64.91       | 71.54      | 50.47 |
> | TDA[9]   | 69.51    | 64.67       | 71.42      | 47.40 |
> | DPE[10]    | 71.91    | 65.44       | 75.07      | 54.20  |
> | TOMCAT | **72.09**   | **64.83**       | **75.46**      | **56.03** |
>
> Table 1. The top-1 classification accuracy on four standard classification datasets.
>
> | Method | UT-Zappos    |            | MIT-States   |            | C-GQA        |            |
> |--------|--------------|------------|--------------|------------|--------------|------------|
> |        | Attribute Acc | Object Acc | Attribute Acc | Object Acc | Attribute Acc | Object Acc |
> | SCEN[5]   | 47.3         | 75.6       | 28.2         | 32.2       | 13.6         | 27.9       |
> | CANet[6]  | 48.4         | 72.6       | 30.2         | 32.6       | 17.5         | 22.3       |
> | CSP[2]    | 47.9        | 73.1       | 38.4        | 54.9      | 30.4        | 44.7      |
> | TOMCAT | **58.5**        | **77.5**      | **41.0**           | **56.6**      | **42.3**        | **54.8**      |
>
> Table 2. The top-1 object classification accuracy on three CZSL datasets (only few works report their classification accuracy).
>
> ---
>
> **Q1:** Explain the compositional zero-shot learning setting.
>
> **R4:** CZSL aims to recognize unseen compositions that have no training data. In other words, the model is trained only on seen compositions during training, but is required to recognize unseen (novel) compositions during testing. **The “zero-shot” is reflected in the fact that the training data of unseen compositions is "zero-shot".**
>
> ---
>
> **Q2:** The impact of ordering of test images.
>
> **R5:** Your question on the test ordering is very insightful. Table 3 provides three different ordering, from which we observe that **there is variance among different orderings, but they are not significant.**
>
> We attribute this to the fact that adaptive update weights enhance the stability of the model, allowing the model to adaptively determine the extent to which the prototypes are updated. In addition, the regularization item of multimodal collaborative representation learning also maintains the stability of the model. Also, we look forward to more future works that can better address this issue.
>
> | Ordering | UT-Zappos |       |       |        | MIT-States |       |       |        | C-GQA |       |       |        |
> |-------|------------|-------|-------|--------|-----------|-------|-------|--------|-------|-------|-------|--------|
> |       | AUC        | HM    | Seen  | Unseen | AUC       | HM    | Seen  | Unseen | AUC   | HM    | Seen  | Unseen |
> | 1     | 48.31      | 60.18 | 74.49 | 72.77  | 22.55     | 39.45 | 50.22 | 52.95  | 11.05 | 28.48  | 37.66  | 34.18   |
> | 2     | 48.52      | 60.82 | 74.64 | 72.95  | 22.46     | 39.33 | 49.84 | 53.05  | 11.38 | 28.93 | 38.44 | 34.18  |
> | 3     | 46.44      | 58.72 | 72.63 | 72.03  | 22.31     | 39.02 | 49.62 | 53.09  | 11.04 | 28.51 | 38.09 | 33.48  |
>
> Table 3. The results of different test ordering on three CZSL datasets.
>
> ---
>
> **Q3:** The impact of the size of label space C.
>
> **R6:** **The size of label space is quite different among the four datasets**: UT-Zappos, MIT-States, C-GQA, and Clothing16K. TOMCAT achieves the state-of-the-art performance on the four datasets, indicating that our method is robust to the size of label space.
>
> In addition, for each dataset, we conduct our experiments **in both closed setting (small label space) and open-world setting (large one)** in Table 1 of the paper. In general, TOMCAT has more obvious advantages over baselines in open-world setting when label space gets larger.
>
> ---
>
> **Q4:** Explain what is the decay.
>
> **R7:** We apologize for only providing a brief description of the decay in the caption of Table 13 in the Appendix G. After the user submits an image, Tomcat first generates a prediction and then performs backpropagation to update the parameters. **Decay denotes the time interval between user’s submission and model’s prediction, including the calculation of logits based on textual and visual prototypes, but excluding the backpropagation.** TOMCAT exhibits minimal decay due to our deliberate design.
>
> ---
>
> Our response to your concerns and questions are provided above. Once again, we appreciate your time and insightful consideration. Please feel free to let us know if you have additional questions.
>
> Best regards,
>
> Authors.
>
> &nbsp;
>
> [1] Jeffrey Pennington, et al. GloVe: Global vectors for word representation. NeurIPS 2014.
>
> [2] Nihal V. Nayak, et al. Learning to compose soft prompts for compositional zero-shot learning. ICLR 2023.
>
> [3] Siteng Huang, et al. Troika: Multi-path cross-modal traction for compositional zero-shot learning. CVPR 2024.
>
> [4] Wentao Bao, et al. Prompting language-informed distribution for compositional zero-shot learning. ECCV 2024.
>
> [5] Xiangyu Li, et al. Siamese contrastive embedding network for compositional zero-shot learning. CVPR 2022.
>
> [6] Qingsheng Wang, et al. Learning conditional attributes for compositional zero-shot learning. CVPR 2023.
>
> [7] Manli Shu, et al.  Test-time prompt tuning for zero-shot generalization in vision-language models. NeurIPS 2022.
>
> [8] Elaine Sui, et al. Just shift it: Test-time prototype shifting for zero-shot generalization with vision-language models. WACV 2025.
>
> [9] Adilbek Karmanov, et al. Efficient test-time adaptation of vision-language models. CVPR 2024.
>
> [10] Ce Zhang, et al. Dual prototype evolving for test-time generalization of vision-language models. NeurIPS 2024.

---

> ### Author Response · Authors · 2025-08-05
> **Response to Reviewer 4ZG6 by Authors**
>
> Dear Reviewer 4ZG6,
>
> We greatly appreciate your time and effort in proposing useful suggestions and being positive to our rebuttal by acknowledgement. To save your valuable time as much as possible, we briefly outline your comments and our corresponding responses:
> - **Confusion by Unclear Writing**：Please rest assured that we will refine our writing and release the code. We would be happy to explain if you point out the unclear points.
> - **Limited Open-world Setting**: We follow all existing open-world CZSL works, sequester the real compositions during training, and let the model infer real and unreal ones automatically at test time.
> - **Bias Towards Compositions**: We report the object accuracy on CZSL datasets, and conduct the experiments to evaluate the performance of our method on four standard classification datasets.
> - **Where is zero-shot reflected**：The “zero-shot” is reflected in the fact that the training data of unseen compositions is "zero-shot".
> - **The impact of test ordering**: The experiments of different ordering shows that the variance is not significant.
> - **The impact of the size of label space**: The label size is quite different among four datasets and two open- and closed-world settings, and our method achieves SOTA performance with robustness.
> - **Explain the Decay**: Decay denotes the time interval between user’s submission and model’s prediction.
>
> Please give us an opportunity to discuss with you if you have any remaining or additional concerns. Once again, thanks for your time and insightful consideration.
>
> With great respect,
>
> Authors.

---

> > ### Comment · Reviewer_4ZG6 · 2025-08-05
> > **Comment**
> >
> > Thanks for your rebuttal. You have addressed my concerns, I have upgraded my score.

---

> > > ### Author Response · Authors · 2025-08-05
> > > **Final Response to Reviewer 4ZG6 by Authors**
> > >
> > > Dear Reviewer 4ZG6,
> > >
> > > Thank you for your careful and thoughtful evaluation of our paper. We greatly appreciate your recognition of our work. We will incorporate these supplementary experiments in our manuscript carefully after rebuttal and release the code after final decision. Thank you again for your valuable suggestions on our paper.
> > >
> > > With our highest respect,
> > >
> > > Authors.

---

### Official Review · Reviewer_hxuG · 2025-07-07

**Clarity:** 4
**Significance:** 2
**Originality:** 3
**Rating:** 4
**Confidence:** 4

**Summary:**

This paper proposes a new method named TOMCAT. By leveraging unlabeled data to accumulate knowledge in both text and visual modalities during the test-time, it aims to address the problem of label space distribution shift caused by unseen combinations in Combinatorial Zero-Shot Learning (CZSL).

**Questions:**

1. **Limited Performance Gain on MIT-States and C-GQA**: Currently, TOMCAT only achieves significant gains on the UT-Zappos dataset (+6.6 AUC for the closed-world setting, +10.7 AUC for the open-world setting), while the gains on the other two datasets are only ≤ 1 AUC. The authors need to provide further explanations for such differences at the data level.​
2. **Add Significance Discussion**: With reference to the statement in Weakness #1, it is recommended to discuss the differences and relationships between the current non-MLLM-based CZSL and MLLM-based attribute recognition/reasoning (e.g., Attribute Recognition and Attribute Reasoning in MMbench [1]) from a more global perspective.

[1] MMBench: Is Your Multi-modal Model an All-around Player? Y Liu and et-al.

**Ethical Concerns:**

["NO or VERY MINOR ethics concerns only"]

**Final Justification:**

The author's response has addressed most of my concerns, I keep positive attitude towards the acceptance of this paper.

**Limitations:**

Yes

**Paper Formatting Concerns:**

No major formatting issues have been found.

**Quality:**

4

**Strengths And Weaknesses:**

**Strength**:

1. **Good Paper Quality**: This work proposes a logically clear and experimentally well-validated solution to the label shift problem in CZSL.
2. **Good Paper Clarity**: The work has a clear structure, rigorous and sufficient arguments, and accurate and appropriate expressions.

**Weakness**:
1. **Limited Significance Regarding Real-world Deployment**: The CZSL paradigm adopted in this paper, when viewed in the context of current powerful MLLMs, has debatable uniqueness and practical significance. For instance, for the task of recognizing a "folded bike", a powerful MLLM can directly perform fine-grained attribute reasoning. In contrast, the setting of CZSL artificially restricts it to a specific scenario where one has seen "folded" and "bike" but not their combination, which deviates from the complexity of the real world.
2. **Limited Originality**: TOMCAT is original at the framework level, but its internal components are inspired by existing works. For example, the idea of TTA has been proven effective in the ZSL field; the dynamic priority queue idea references TDA; COOP is used during the training phase; the fusion method for prediction references Tip-adapter; and the design of KAM may also refer to the prototype shift idea in TPS. Although the authors have clearly cited most of these references, the overall originality is somewhat limited.

---

> ### Author Rebuttal · Authors · 2025-07-30
>
> Dear Reviewer hxuG,
>
> We sincerely and deeply thank you for your valuable reviews, which truly contribute a lot to improving the quality of our paper. First, we appreciate the recognition of our paper: (1) TOMCAT is a logically clear and experimentally well-validated solution for CZSL; (2) Our work is well-structured, well-argued, and clearly expressed. Below, we address your questions and clarify the misunderstanding.
>
> ---
>
> **W1:** The significance of CZSL in the context of current powerful MLLMs.
>
> **R1:** At its core, CZSL is a **representation learning task** rather than a **generative task**; hence, MLLMs are not suitable for this purpose, for the following reasons:
>
>
> 1. In the current situation where the community pays excessive attention to generative models such as LLMs and MLLMs, representation models like CLIP have the advantages of **strong feature extraction ability**.
> 2. MLLMs are used for text generation, so **their model behavior is difficult to control**.
> 3. MLLMs have billions of parameters, whose inference **needs too much GPU memory and time**. CLIP based methods are more flexible under resource-constrained and latency-sensitive conditions.
> 4. As shown in Table 1, we conduct the experiments on the three datasets in CZSL using MLLMs (e.g., LLaVA v1.5[1], and InternVL-3[2]). **Our TOMCAT outperforms the MLLMs** significantly, indicating that MLLMs, whose generation is hard to control, are not suitable for representational tasks like CZSL, recognition, and retrieval.
>
> | Method     | UT-Zappos     |             |                  |            | MIT-States |             |                  |       |
> |------------|---------------|-------------|------------------|------------|---------------------|-------------|------------------|-------|
> |            | Attribute Acc↑ | Object Acc↑ | Composition Acc↑ | Time (ms)↓ | Attribute Acc↑       | Object Acc↑ | Composition Acc↑ | Time (ms)↓ |
> | LLaVA v.15[1] | 2.13          | 10.17       | 0.32             | 463        | 1.06                | 21.77       | 5.72             | 857   |
> | InternVL-3[2] | 8.77          | 23.71       | 2.06             | 5770       | 8.53                | 48.28       | 4.43             | 728   |
> | TOMCAT     | **58.54**         | **77.45**       | **44.88**            | **47**        | **41.00**                  | **56.58**       | **30.92**            | **145**   |
>
> Table 1. The classification accuracy of TOMCAT and two MLLMs. Since MLLMs cannot generate classification probability, we report the top-1 accuracy instead of AUC and HM.
>
> 5. **CZSL has broad applicability**, such as enabling home robots to recognize novel attribute-object compositions like “broken glass” or “dirty plate” for accurate classification; and facilitating drug discovery by composing different chemical molecules to predict bioactivity.
> 6. For the sake of research convenience, CZSL simplifies compositions into single-attribute single-object pairs. However, CZSL can **easily generalize to multi-attribute single-object**, encoded by CLIP text encoder, such as "old and folded bike".  Some recent works, such as Trident[3] and MAC[4], have started to address this aspect.
>
> ---
>
> **W2:** TOMCAT has limited originality.
>
> **R2:** We appreciate the reviewer’s concern regarding the originality of our work, but we would like to clarify our motivation more clearly:
>
> 1. Our work is **the first to pay attention to the issue of label space shift in CZSL**, and for the first time use unsupervised test data to adapt to the new label distribution at test time, improving performance on both seen and unseen categories. The entropy loss used during test overcomes the drawback that unseen compositions can not be included for optimization during the training stage. We believe that our method, from the angle of label space shift to solve the CZSL problem, will provide significant inspiration for future research in the CZSL community.
> 2. Our method is inspired by TDA[5], TPS[6], and DPE[7]. **However, TOMCAT is different from them at the methodology level**. TDA records historical images without focusing text information; TPS concentrates on shift vectors but fails to use visual information; DPE updates all prototypes by directly adding vectors to them **in a offline manner** (**failing to accumulate knowledge**), which treats all categories equally regardless of familiarity.
> Our work overcomes their limitations with visual priority queue, **online** knowledge accumulation modules, and **adaptive update weights** based on the similarity of test image and multi-modal prototypes. **Our work is quite different from the above ones, so we believe it still has considerable originality.**
>
> ---
>
> **Q1:** Explanation on the varying degree of improvement on datasets.
>
> **R3:** Your observation on the phenomenon is very perceptive, which we have noticed when conducting experiments and briefly mentioned in Sec.4.2 Main Results.
>
> 1. MIT-States and C-GQA are two datasets containing real-world and daily compositions (e.g., tiny dog). These datasets share a similar image style with the CLIP pre-training datasets, which determines that the CZSL **training** set is good to fine-tune CLIP for the CZSL task, leaving limited room for improvement through **test-time adaptation**.
> 2. However, UT-Zappos and Clothing16K are fine-grained fashion datasets, whose image style and label space shift are very different from CLIP pre-training datasets. At this point, the CZSL training set can only adapt CLIP to the seen compositions in CZSL, without enabling it to learn knowledge about the unseen ones. TOMCAT has great potential in learning unseen compositions as time goes by.
>
> **In conclusion, The greater the label shift in the CZSL dataset, the more obvious the advantage of our method over other CZSL methods.**
>
> ---
>
> **Q2:** The difference between CLIP-based CZSL and MLLM-based attribute recognition.
>
> **R4:** The significance of CZSL is responded in **R1**. Here we pay more attention to the difference between the two settings. **As general-purpose models, the attribute recognition capability of MLLMs is limited to coarse-grained scenarios. However, CZSL focuses on fine-grained compositional reasoning under zero-shot setting (a kind of representational task).**
>
> As shown in Table 1, for more fine-grained dataset UT-Zappos and Clothing16K, MLLMs can not systematically classify the compositions well. Moreover, fine-tuning MLLMs on specific datasets is challenging due to the limited data size, label format constraints, and memory requirements. Instead, our CLIP-based TOMCAT performs well with less time usage.
>
> ---
>
> The above is my response to the question and concern raised. Once again, thank you for your time and thoughtful consideration. We would be happy to discuss any remaining concerns in the next phase.
>
> Best regards,
>
> Authors.
>
> &nbsp;
>
> [1] Haotian Liu, et al. Improved baselines with visual instruction tuning. CVPR 2024.
>
> [2] Jinguo Zhu, et al. Internvl3: Exploring advanced training and test-time recipes for open-source multimodal models. arXiv preprint 2025.
>
> [3] Xudong Yan, et al. Leveraging MLLM Embeddings and Attribute Smoothing for Compositional Zero-Shot Learning. arXiv preprint 2024.
>
> [4] Shuo Xu, et al. MAC: A Benchmark for Multiple Attributes Compositional Zero-Shot Learning. arXiv preprint 2024.
>
> [5] Adilbek Karmanov, et al. Efficient test-time adaptation of vision-language models. CVPR 2024.
>
> [6] Elaine Sui, et al. Just shift it: Test-time prototype shifting for zero-shot generalization with vision-language models. WACV 2025.
>
> [7] Ce Zhang, et al. Dual prototype evolving for test-time generalization of vision-language models. NeurIPS 2024.

---

> ### Author Response · Authors · 2025-08-06
> **Author Response to Reviewer hxuG**
>
> Dear Reviewer hxuG,
>
> We are sincerely grateful for your careful evaluation and constructive suggestions. We know the reviewers are busy and their time is valuable, so we briefly outline the comments and the corresponding response to save your time:
>
> - **Limited Significance of CZSL under MLLM background**: CZSL is a representational learning task, instead of generative task. And the the results show MLLM-based methods are less effective than CLIP-based methods in CZSL.
> - **Explanation on Originality**: Our motivation is to pay attention to the issue of label space shift in CZSL, and use unsupervised test data to adapt to the new label distribution at test time, which represents a novel direction not previously considered. **We believe our work, like CSP--which has made CLIP model and prompt tuning become mainstream techniques for CZSL--will inspire a new wave of test-time adaptation based approaches.**
> - **Varying degree of improvement on Datasets**: UT-Zappos and Clothing16K datasets have greater label shift, which better boosts the performance of our model.
>
> We would be very grateful for the opportunity to discuss any remaining or additional concerns that you may have. Thank you again for your time and thoughtful consideration.
>
> With great respect,
>
> Authors.

---

### Comment · Area_Chair_DbzN · 2025-08-02

Dear all,

Thanks for your engagement in the review process.

***Please read all other reviews and the author responses carefully, and provide your response as soon as possible***. The Author-Reviewer Discussions (July 31 - Aug 6) are crucial for ensuring review quality and providing authors with an opportunity to address questions and potential misunderstandings from reviewers. Please treat it seriously.

Again, thanks for your work so far, and ***please do participate in the Author-Reviewer Discussions ASAP***.

Best,

AC

---

### Author Response · Authors · 2025-08-04
**Author Response**

Dear AC and Reviewer hxuG, 4ZG6, uhct, 45LQ, and PjJh,

Thank you for your time and insightful comments. We know all reviewers’ time is valuable; however, the author-reviewer discussion period is more than halfway over. We have tried our best to conduct the experiments and provide detailed response to address each reviewer’s concerns.

We sincerely ask if you could kindly take a moment to briefly read our response. And we would be very happy to discuss with you if there are any remaining concerns.

With sincere regards,

Authors.

---

### Decision · Program_Chairs · 2025-09-17

**Decision:**

Accept (poster)

**Comment:**

The work addresses a meaningful challenge in CZSL, i.e., the distribution shift caused by unseen attribute-object compositions at test time. The authors propose a novel method that leverages unlabeled test data to dynamically update multimodal prototypes, incorporating an adaptive weighting mechanism and a priority queue to accumulate knowledge. This approach is justified as a way to adapt to the shifting label space without ground-truth labels.

The paper's strengths include its good empirical performance on standard CZSL benchmarks under both closed and open-world settings. The comprehensive evaluation and clear presentation are notable. Some concerns were raised regarding the fairness of comparison to methods that do not use test-time data, the scalability of the approach, and the degree of methodological novelty relative to existing test-time adaptation techniques. The authors' rebuttal has provided additional experiments comparing their method to MLLMs and other TTA techniques, demonstrated robustness to test-time ordering, and included results on standard classification datasets to show broader applicability. These responses adequately addressed most reviewer concerns, leading several reviewers to raise their scores.

Given the above conditions, I recommend an acceptance.